

# Information Content Analysis: The Potential for Methane Isotopologue Retrieval from GOSAT-2

Edward Malina[1], Yukio Yoshida[2], Tsuneo Matsunaga[2], Jan-Peter Muller[1]

[1]Imaging Group, Mullard Space Science Laboratory, Department of Space and Climate Physics, University College London, Holmbury St. Mary, Dorking, Surrey, RH5 6NT, UK;
[2]Center for Global Environmental Research/Satellite Observation Center, National Institute for Environmental Studies, 16-2 Onogawa, Tsukuba, Ibaraki, JAPAN 305-8506

*Correspondence to*: Edward Malina (edward.malina.13@ucl.ac.uk)

**Abstract.** Atmospheric methane is comprised of multiple isotopic molecules, with the most abundant being $^{12}CH_4$ and $^{13}CH_4$ making up 98% and 1.1% of atmospheric methane respectively. It has been shown that is it possible to distinguish between sources of methane (biogenic methane, e.g. marshland or abiogenic methane, e.g. fracking) via a ratio of these main methane isotopologues, otherwise known as the $\delta^{13}C$ value. $\delta^{13}C$ values typically range between -10 and -80 per mil, with abiogenic sources closer to zero, and biogenic sources showing more negative values. Initially, we suggest that a $\delta^{13}C$ difference of 10 per mil is sufficient, in order to differentiate between methane source types, based on this we derive that a precision of 0.25ppbv on $^{13}CH_4$ retrievals may achieve the target $\delta^{13}C$ variance. Using an application of the well established Information Content Analysis (ICA) technique for assumed clear sky conditions, this manuscript shows that using a combination of the Shortwave Infrared (SWIR) bands on the planned Greenhouse gases Observing SATellite (GOSAT)-2 mission, $^{13}CH_4$ can be measured with sufficient Information Content (IC) to a precision of between 0.7 and 1.2ppbv from a single sounding (assuming a total column average value of 19.14ppbv), which can then be reduced to the target precision through spatial and temporal averaging techniques. We, therefore, suggest that GOSAT-2 can be used to differentiate between methane source types. Large unconstrained 'a priori' covariance matrices are required in order to achieve sufficient information content and that varying the solar inclination angle has limited impact on information content or retrieval errors.

## 1 Introduction

Of the major Greenhouse Gases (GHGs) currently considered as having a major impact on atmospheric chemistry, methane is amongst the most important. The potential for atmospheric heating by methane is well documented (IPCC, 2014; Khalii and Rasmussen, 1994; Kirschke et al., 2013; Wuebbles and Hayhoe, 2002). Excess concentrations of atmospheric methane can lead to detrimental effects on the chemistry of the atmosphere, as well as the absorption of Infra-red (IR) radiation causing atmospheric heating. Methane concentration in the atmosphere has been documented to be rising steadily over the past century, aside from a short period in the middle of the last decade (Heimann, 2011; Kai et al., 2011), leading to renewed





efforts in understanding global atmospheric methane. In order to tackle the problem of growing methane concentrations, it is necessary to understand the nature of the global sources of methane that will allow for a greater understanding of the processes behind methane generation and how they will affect the global environment. A key point is that the global methane budget is still not truly understood, this is highlighted by the "pause" in the increase in global methane concentration in the

last decade, where there are many contrasting arguments published explaining its cause (Aydin et al., 2011; Heimann, 2011; Kai et al., 2011). Towards this end of understanding global methane emissions (and other Greenhouse Gases (GHGs)), multiple satellite missions have been launched, including the SCanning Imaging Absorption spectroMeter for Atmospheric CHartographY (SCIAMACHY) (Bovensmann et al., 1999) and the Greenhouse gases Observing SATellite (GOSAT) (Kuze et al., 2009), with future GHG monitoring missions currently under development.

What is suggested is that there may be profound disagreement as to whether the majority of atmospheric methane occurs from natural or anthropogenic sources, and satellite measurements to date have not yet addressed this problem. Towards this end, we consider an assessment of the potential of measuring the main isotopologues of methane (methane consisting of different carbon and/or hydrogen isotopes) from a spaceborne instrument. Atmospheric methane is primarily composed of

two key isotopologues, $^{12}CH_4$ and $^{13}CH_4$, which have a natural abundance of about 98% and 1.1% respectively. It is a well-established fact that different sources of methane (i.e. biogenic sources such as methanogens and Arctic permafrost (EPA, 2010), or non-biogenic such as industrial hydrocarbon burning) vary in the abundance of these isotopologues (Etiope, 2009; Rigby et al., 2012). This fractionation between sources generally occurs for two reasons: (1). Plant based photosynthesis enzymes discriminate against carbon dioxide during uptake, because of the higher isotopic mass, thus most plant-based

material are depleted in $^{13}C$ hydrocarbons (2). The bacterial reduction of carbon dioxide to methane is associated with a Kinetic Isotope Effect (KIE), which discriminates against $^{13}C$, thus leaving depleted $^{13}CH_4$ concentrations in biogenic methane sources (Levin et al., 1993; Whiticar, 1999), this nominally occurs in soil and therefore is not associated with the other forms of methane (thermogenic and abiogenic). Comprehensive reviews of these discriminations and global sources of such are reviewed in more detail elsewhere (Nisbet et al., 2016; Schaefer et al., 2016; Schwietzke et al., 2016).

Methane isotopologue measurements of a sample of air are typically expressed as per mil ratios of the heavier to lighter isotopologues relative to an established literature standard, which in the case of $^{13}C$ to $^{12}C$ is Vienna Pee Dee Belemnite (VPDB).

$$\delta^{13}C = \left( \frac{\left(\frac{13C}{12C}\right)sample}{\left(\frac{13C}{12C}\right)standard} - 1 \right) \times 1000 \quad (1)$$

In the case of this work, we assume all measurements are in the form of total column averaged values. Note that VPDB is unusually enriched in $^{13}C$ methane, meaning that all measurements taken in reference to VPDB will most likely have negative values. Due to the reasons stated above, biogenic sources of methane will have $\delta^{13}C$ values in the range of -60 to -80 per mil, while industrial sources should have values closer to -40 or -30 per mil (Rigby et al., 2012).





Unlike methane, global measurements of the $\delta^{13}C$ ratio are much more limited, with most of the publically available data restricted to 20 sites of in-situ measurements from the National Oceanic and Atmospheric Administration (NOAA) carbon cycle greenhouse gas cooperative air sampling network (http://www.esrl.noaa.gov/gmd/ccgg/flask.html). Although these
measurements are extremely accurate and useful in their own right, they are limited by their sparseness and physical location. They are all in areas which sample background values rather than anomalies associated with large methane sources. This means that they can only provide limited guidance on global distributions of $\delta^{13}C$. Some measurements from balloon soundings (Röckmann et al., 2011) and satellite based solar occultation measurements (Buzan et al., 2016) are available, but these only sample the atmosphere from the mid to the upper troposphere where methane is well mixed and is in contact with
Hydroxyl which is likely to destroy the original isotopologue signature and so miss the key activity, which occurs in the lower troposphere which is of most interest to the scientific community. Such lower tropospheric activity can only be captured from a satellite instrument with a nadir sounding profile, preferably in the Shortwave Infrared (SWIR). Therefore, if total column soundings of $\delta^{13}C$ can be retrieved from a satellite platform and yield enough information with a sufficiently high degree of precision, there is potential for very useful information on the global distribution of biogenic and non-
biogenic methane sources. Rigby et al (2012) suggest that there is a minimum margin of 10 per mil in terms of differentiating between fossil fuel and biogenic sources. Using Eq. (1), we can estimate the minimum precision on $^{13}C$ methane measurements required to achieve this per mil margin, on the assumption that $^{12}C$ methane measurements have a precision of 6ppbv (Yoshida et al., 2011). Applying Eq. (1) the minimum precision is suggested to be 0.25ppbv on the $^{13}C$ retrievals for a $\delta^{13}C$ range of 10 per mil. If the precision is lower than this, then we lose the ability to differentiate between
source types for certain, but it may still be possible if the measured $\delta^{13}C$ values are at the extreme end of the scale.

Analysis of global $\delta^{13}C$ concentrations by Nisbet et al. (2016) shows that trends and variations in $\delta^{13}C$ on a regional scale are of the order of a few per mil, which suggests that any total column retrieval algorithm will have to obtain better than this precision in order to comment on trends in $\delta^{13}C$. Given the above assessment, a much more likely prospect is the analysis of
localised regions, where Nisbet et al. (2016) state that they can see wider ranges in the $\delta^{13}C$ of different source regions, for example, Arctic and boreal wetland regions showing a per mil value of -70, while Siberian gas fields are at the -50 mark.

The fact that we are mostly interested in lower tropospheric sources of methane makes satellite measurements in the SWIR band much more useful than in the Thermal Infrared (TIR) due to higher surface sensitivity (Herbin et al., 2013; Worden et
al., 2007). Most current and future SWIR nadir satellites assume a passive solar-surface-satellite light path, which is largely the cause of higher surface sensitivity for these instruments. This statement is built on the assumption of a significant number of methane isotopologues spectral lines present in the SWIR, and in the sensitivity ranges of any instruments. However, this particular passive remote sensing method is highly susceptible to light path modification due to aerosols and clouds, thus adding high degrees of uncertainty to the retrieval, yet because in this assessment we are interested in the ratio of two gases





($^{12}CH_4$ & $^{13}CH_4$), we can apply the proxy method (Frankenberg et al., 2011; Parker et al., 2011), which assumes that the light path modification of two spectrally close traces gases will be similar, and will, therefore, cancel out when calculating a ratio. The High Resolution Transmission (HITRAN) 2012 database (Rothman et al., 2013) states that there are multiple methane isotopologues absorption features present in the 1600-1700nm and 2200-2300nm wavebands. Both of these wavebands are

included in the sensitivity range of the planned GOSAT-2/TANSO-FTS-2 instrument (GOSAT-2 Project Team, 2015). Therefore in order to maximise the potential quantity of information available to a given instrument, this work focuses on the degree of information available in the GOSAT-2 sensitivity range. Although the primary goal of this work is to investigate GOSAT-2, the 1600-1700nm waveband is also present in the current GOSAT/TANSO-FTS instrument, therefore any investigations into this waveband with GOSAT-2 are also likely to be applicable to GOSAT.

In this work, we apply the well-established Information Content Analysis (ICA) techniques originally proposed by Rodgers (2000) to determine the potential benefit of retrieving total column methane isotopologue concentrations using bands 2 and 3 of the GOSAT-2/TANSO-FTS-2 instrument. The value of such studies has been proven on multiple occasions (Frankenberg et al., 2012; Herbin et al., 2013; Kuai et al., 2010; Yoshida et al., 2011), providing guidance on appropriate potential retrieval

setups in order to maximise information received from trace gas retrievals. The original Optimal Estimation Method (OEM) proposed by Rodgers (2000) generally requires 'a priori' knowledge of the retrieval setup. However, due to the fact that there has been limited research in this area and no 'a priori' state vectors or Variance Covariance Matrices (VCMs) have been defined previously, we test a number VCMs in order to explore the constraints on retrieving independent information in the total column based on the ICA. This analysis and VCM variations also provide an opportunity to explore the potential

errors associated with retrievals of isotopologues in these wavebands (Ceccherini and Ridolfi, 2009; Yoshida et al., 2011).

## 2 Models and Instruments

In the following subsection, we describe the tools and assumed instruments employed during the course of this research.

### 2.1 The Oxford Reference Forward Model (ORFM)

The ORFM (Dudhia, 2017) is a General Line by Line (GENLN2) based Radiative Transfer Model (RTM) originally developed at the University of Oxford to provide reference spectral calculations, for the Michelson Interferometer for Passive Atmospheric Sounding (MIPAS) instrument based on the ENVISAT satellite. Multiple viewing geometries are possible, as well as advanced physical effects such as water vapour continuums and carbon dioxide line mixing. Outputs from the ORFM include transmission, absorption, radiance, optical depth and brightness temperature, making the ORFM a

highly versatile tool. The ORFM is a popular RTM used within the National Centre for Earth Observation (NCEO) community in the United Kingdom (UK) and has trace gas retrieval heritage (Illingworth et al., 2014). The ORFM does not



currently include an illumination source such as a 'sun', so cannot generate SWIR radiance spectra 'out of the box'. Instead, we generate SWIR radiance spectra by multiplying transmission spectra generated in the ORFM with a reference solar irradiance spectrum, namely the Committee on Earth Observation Satellites – Working Group on Calibration and Validation (CEOS-WGCV) recommended SOLar SPECtrum (SOLSPEC) (Thuillier et al., 2003).

**2.2 GOSAT-2**

GOSAT-2 is due to be launched in the 2018 financial year of Japan and is a follow on from the original GOSAT mission launched in 2009. GOSAT-2 like GOSAT is a collaborative effort between the Ministry of the Environment (MOE), the

10 Japan Aerospace Exploration Agency (JAXA) and the National Institute for Environmental Studies (NIES) in Japan. GOSAT-2 aims to continue the legacy of GOSAT by providing global measurements of methane and carbon dioxide in order to monitor GHG emissions, as well as provide new scientific data focusing on localised flux and point source emissions. GOSAT has an established history of providing reliable methane products (Parker et al., 2011; Schepers et al., 2012; Yoshida et al., 2011). GOSAT-2 represents one of the best opportunities for measuring methane isotopologues with this new

generation of GHG satellite instruments. With the combination of the GOSAT and GOSAT-2 satellites, there will be a nearly unbroken record of global GHG emissions between 2009 and 2022 (with a 5 year lifetime planned for GOSAT-2), providing an unprecedented record on GHG emissions. The TANSO-FTS-2 instrument is similar to the TANSO-FTS instrument (Kuze et al., 2009) but, in the context of this study, has a significant advantage, which is the extension of band 3 up to 2380 nm, where significant numbers of methane spectral lines are located (Table 1). Therefore, this study focuses on the original

GOSAT SWIR sensitivity region of 1560-1690 nm (band 2, also present in GOSAT-2) and the new SWIR sensitivity band in order to maximise any potential information on methane isotopologues. The exact technical details of GOSAT-2 are not yet available, however, due to the similarity of the instruments, we assume that the Signal to Noise Ratio (SNR) and Instrument Line Shape Function (ILSF) on GOSAT-2/TANSO-FTS-2 and GOSAT-TANSO-FTS are similar, and are explained in more detail below (GOSAT-2 Project Team, 2016).

Table 1. Spectral coverage of TANSO-FTS and TANSO-FTS-2.

|  | *Band 1* | *Band 2* | *Band 3* | *Band 4* | *Band 5* |
|---|---|---|---|---|---|
| *TANSO-FTS* | 0.76 ~ 0.78 μm | 1.56 ~ 1.72 μm | 1.92 ~ 2.08 μm | 5.5 ~ 14.3 μm | N/A |
|  | 12900 ~ 13200 cm$^{-1}$ | 5800 ~ 6400 cm$^{-1}$ | 4800 ~ 5200 cm$^{-1}$ | 700 ~ 1800 cm$^{-1}$ |  |
| *TANSO-FTS-2* | 0.75 ~ 0.77 μm | 1.56 ~ 1.69 μm | 1.92 ~ 2.38 μm | 5.5 ~ 8.4 μm | 8.4 ~ 14.3 μm |
|  | 12950 ~ 13250 cm$^{-1}$ | 5900 ~ 6400 cm$^{-1}$ | 4200 ~ 5200 cm$^{-1}$ | 1188 ~ 1800 cm$^{-1}$ | 700 ~ 1188 cm$^{-1}$ |





### 3 Information Content Analysis - Theory

In order to identify the figures of merit that will be used for ICA in this study, we must first briefly outline the theory behind OEM. OEM theory was originally published by Rodgers (2000), and in the case of this study, we use Yoshida et al's (2011) interpretation. However, in the case of ICA, there is no retrieval step included since we make the assumption of evaluating

5   the ICA at the linearization point (the 'a priori' vector).

OEM theory is fundamentally based on the estimation of the state vector (atmospheric profile) **x** given a set of measurements **y**. This relationship is typically expressed as:

$$y = F(x, b) + \varepsilon \quad (2)$$

where **F(x,b)** is the forward model relating the atmospheric state to the measurements, **b** is a model parameter vector necessary for computations but is not retrieved and **ε** is an error vector comprising forward model errors and instrument errors. The optimal estimate (solution) for **x** using a non-linear maximum a posteriori (MAP) method is achieved through minimising the cost function:

$$J(x) = [y - F(x, b)]^T S_\varepsilon^{-1} [y - F(x, b)] + (x - x_a)^T S_a^{-1} (x - x_a) \quad (3)$$

where $x_a$ is the 'a priori' state of **x**, $S_a$ is the VCM about the 'a priori' state, and $S_\varepsilon$ is the error covariance matrix. However, since this section of the analysis does not include a retrieval step, we can linearly solve Eq. (3) as follows:

$$x = Gy + (I - GK)x_a \quad (4)$$

where **K** is the Jacobian matrix (or weighting function), defined as the derivative of the forward model as a function of the state vector, and is quantitatively defined as $K = \partial F(x, b)/\partial x$. The Jacobian matrix effectively describes the sensitivity of the forward model to changes in the state vector. **G** represents the Gain matrix, which describes the sensitivity of the final retrieved state vector to changes in the measurements, it is quantitatively described as:

$$G = S_{a,x} K_x^T \left( S_\varepsilon + K_x S_{a,x} K_x^T + K_c S_{a,c} K_c^T \right)^{-1} \quad (5)$$

where the subscripts $x$ and $c$ refer to sub-matrices for target species (in this case $^{13}CH_4$) and auxiliary/interfering elements respectively. Using these relationships we can define an information quantity, the 'Averaging Kernel' as:



$$\frac{\partial \hat{x}}{\partial x} = A = GK_x \quad (6)$$

where $\hat{x}$ is the a posteriori estimate of the state vector. The Averaging Kernel quantitatively describes the sensitivity of the final retrieved state vector to changes in the true state vector. In other words, in the context of this study, if we assume the true state vector is the 'a priori' state, then the Averaging Kernel describes the ability of the retrieval to infer deviations in

state vector elements away from the 'a priori' state. Thus if **A** was an identity matrix it would represent a perfect retrieval, since all elements of the state vector would reproduce any changes with no interference. Given this fact the trace of **A** indicates the number of independent pieces of information a retrieval provides, otherwise known as the Degrees of Freedom for Signals (DOFS), quantitatively described:

$$DOFS = trace(A) \quad (7)$$

Thus in order to obtain relevant information out of a retrieval, the DOFS value must be greater than or equal to unity with each diagonal element of the averaging kernel representing a partial degree of freedom attached to a specific atmospheric layer, for a specific atmospheric parameter. The averaging kernel does not provide information on the expected errors in the $^{13}CH_4$ channels, therefore we must definite a total error covariance matrix of the 'retrieval state'. The total error covariance is defined as the sum of the measurement noise $S_m$, smoothing error $S_s$ and interference error $S_i$, each of these quantities are

defined below:

$$S_m = G_x S_\varepsilon G_x^T \qquad (8)$$
$$S_s = (A_{xx} - I) S_{e,x} (A_{xx} - I)^T \qquad (9)$$
$$S_i = A_{xc} S_{e,c} A_{xc}^T \qquad (10)$$

where $S_e$ is an ensemble 'a priori' covariance matrix and the subscripts $x$ and $c$ denote the sub-matrices for target gases or auxiliary elements respectively.

The impact of these covariances is indicated in Yoshida et al. (2011) for $^{12}CH_4$, the main methane molecule, where measurement and smoothing error form the main components of the error. The impact of the errors on any potential

retrievals on $^{13}CH_4$ is discussed below.

**4. 'a priori' set up and covariance composition**

Making use of the ORFM, simulated unpolarised SWIR radiance spectra are generated based on an atmospheric model created at the University of Leicester for operational processing of the MIPAS instrument. The model provides a high level of vertical resolution and gas concentrations at 2002 estimates. This model is used throughout this paper and is designed to





simulate mid-latitude daytime conditions. The model does not have concentration values for $^{13}CH_4$, therefore a profile was generated based on the HITRAN $^{13}C/^{12}C$ ratio, which is 1.11031% of the methane column. This paper generates vertical 'a priori' state vectors based on this model, assuming a 21 level atmosphere between 0 and 63km, with a high density in representation in the troposphere, and sparse representation (2-3) in the stratosphere, since SWIR are far more sensitive

nearer the surface. It should be noted that Yoshida et al. (2011) use 15 atmospheric levels, and Parker et al. (2011) employ 20.

### 4.1. 'A priori' and its error covariance

The 'a priori' error covariance matrix can be generated based on transport models such as the NIES TM (Saeki et al., 2013) or from in-situ data such as from the TCCON network. There are many examples of appropriate covariance matrices for the

purpose of GOSAT based trace gas measurements (Eguchi et al., 2010; Yoshida et al., 2011), however there are no examples of $^{13}CH_4$ 'a priori' error covariance matrices in the established literature, nor are there any transport models that can provide reliable values at this time. It was, therefore, necessary to experiment with a number of matrices, in order to establish a covariance matrix that would provide sufficient information on the GOSAT-2 channels. The starting point for these matrices is based on the assumption that the maximum variations on $\delta^{13}C$ that are likely to be observed, ranging from -10 to -80 per

mil (Rigby et al., 2012; Sherwood et al., 2016). Therefore, we can assume that the average global variation of $\delta^{13}C$ is -45 +/- 35 per mil. Applying Eq. (1), we can determine that a per mil variation of 35 equates to roughly $(3\%)^2$ variance in the $^{13}CH_4$. This is a low variance for a minor species, and is unlikely to yield any information. Therefore, this study initially assumes a $(10\%)^2$ variance.

This study defines two forms of the matrix, firstly a pure diagonal covariance matrix based on the equation:

$$\boldsymbol{S}_{a,ii} = \sigma_{a,i}^2 f^2 \qquad (11)$$

where $\mathbf{S}_{a,ii}$ is element $i,i$, (atmospheric layer) of a diagonal matrix, $\sigma_{a,i}$ is the standard deviation of element $i$ of the 'a priori' vector, which in the case of this assessment is initially set at $(10\%)^2$, and $f$ is a scaling factor designed to increase or decrease

the standard deviation of the elements of the covariance matrix. This factor $f$ is designed to determine at what point the inherent instrument noise no longer has any influence on the retrieval. Because $^{13}CH_4$ is present in minimal quantities in the atmosphere, it was deemed necessary to explore the effects of a non-diagonal covariance matrix, where the off-diagonal elements are calculated using the equation (Illingsworth et al., 2014):

$$\boldsymbol{S}_{a,ij} = \sqrt{S_{a,ii}S_{a,jj}}exp\left(\frac{-(z_i-z_j)^2}{z_S^2}\right) \qquad (12)$$





where $S_{a,ij}$ refers to a given off-diagonal element of layer $i,j$, $z_i$ is the altitude of element $i$, $z_j$ is the altitude of element $j$ and $z_S$ is the smoothing length, nominally set between 1 and 3km.

The other key gases present in bands 2 and 3 of GOSAT-2 ($^{12}CH_4$, $CO_2$, $H_2O$ and $CO$) are all set at $(10\%)^2$ of the MIPAS atmospheric profile, matching the initial value of the $^{13}CH_4$ covariance matrix, with Herbin et al. (2013) suggesting similar

5  variations at their peak.

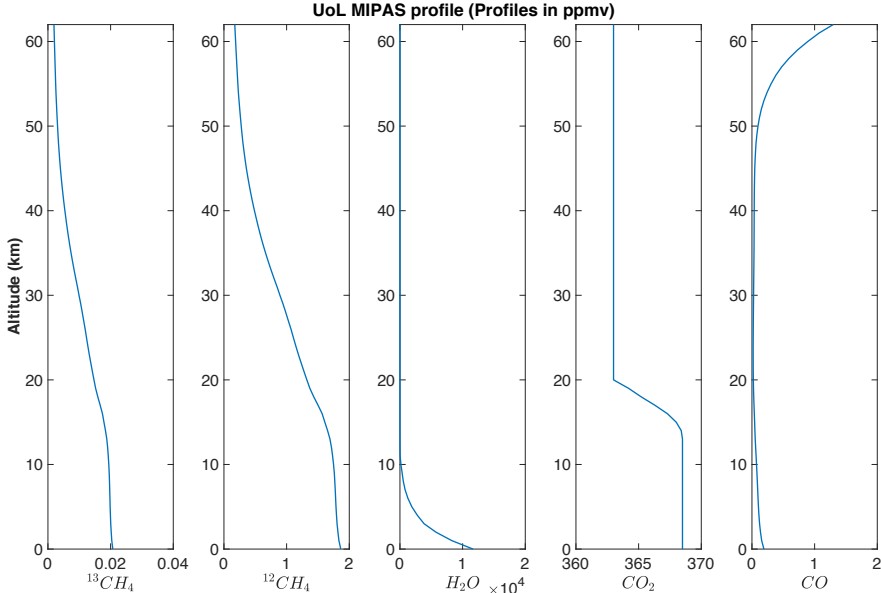

**Figure 1. 'A priori' gas concentration profiles of the main gases of interest.**

The MIPAS model assumes a total column averaged methane concentration of 1740ppbv, which assuming a $^{13}C$ ratio of 1.1%, equates to a total column averaged concentration of 19.14ppbv for $^{13}CH_4$.

The information content analysis 'a priori setups', and simulation set ups are summarised in the table below following the style of Herbin et al (2013):





**Table 2. Parameters for Information Content Analysis**

| State vector elements | $^{12}CH_4$ | $^{13}CH_4$ | $H_2O$ | $CO_2$ | $CO$ | Surface Albedo |
|---|---|---|---|---|---|---|
| A priori values ($x_a$) | | | As Model MIPAS Atmosphere | | | Between 0.1 and 0.6 simulating vegetation and desert conditions |
| Covariance Standard Deviation | 10% | 10-100% | 10% | 10% | 10% | Not assessed |

Variations are based on those values shown by Eguchi et al. (2010) and Herbin et al. (2013), but at their maximum with the aim of determining the maximum interference error for $^{13}CH_4$ for maximum DOFS.

### 4.2. Measurement Error covariance matrix

Instrument performance is a crucial component of any ICA, however, in the case of GOSAT-2/TANSO-FTS-2, the exact details of the FTS performance are not yet published. Therefore, instrument line shape functions and SNR values equivalent
to GOSAT/TANSO-FTS are assumed, purely for the SWIR bands. The assumed SNR is a factor of multiple components of the instrument, and in the SWIR is a combination of inherent instrument noise (dark current) and noise from received photons (shot noise). Assuming varying land surface types and solar inclination/satellite inclination angles (from vegetation to desert), GOSAT has an SNR range between 300 and 500 (Yoshida et al., 2011) over band 2 (potentially lower over water surfaces), and for the purposes of this study similar SNR values are assumed for band 3 taking into account the lower
radiance values of band 3. Based on this knowledge, the instrument error covariance matrix is defined as:

$$S_{\varepsilon,ii} = \sigma_{\varepsilon,i}^2 \qquad (13)$$
$$S_{\varepsilon,ij} = 0 \qquad (14)$$

where:

$$\sigma_{\varepsilon,i} = \frac{\frac{\sum_{j=0}^{n} y}{n}}{SNR} \qquad (15)$$



where $\sigma_{\varepsilon,i}$ is the standard deviation of the $i$-th measurement of the measurement vector **y.** The diagonal values of the covariance matrix are identical since the SNR is applied to the entirety of the measurement bands, rather than individual measurement values.

We note that additional errors can be incorporated into the measurement error covariance matrix, most notably errors from the forward model, however, in the case of this study, we assume that the majority of the errors are to be found in the instrument and forward model errors are not important. In the case of a full retrieval, forward model errors must be accounted for in order to make accurate measurements, however in the case of information content determination we are justified in ignoring them as they will not impact the information content to any significant degree (Frankenberg et al., 2012; Herbin et al., 2013).

## 4.3. Non-retrieved elements

The complexity of atmospheric retrievals requires that we consider the potential impact of elements outside the main retrieval parameters (**b** in Eq. (2)). These quantities have not been included in the equations identified above since it is beyond the scope of this work, but the potential effects are described in detail in this section. Yoshida et al. (2011) lay particular emphasis on potential instrumentation effects (outside those contained within the SNR) that are important to include in the retrieval vector, such as the wavenumber dispersion (an effect of self-apodization and other effects). However, although we expect such effects to be present in TANSO-FTS-2, we judge that they are not important in the context of information content and are therefore not further considered in this work.

As highlighted in Yoshida et al. (2011), the GOSAT SWIR channel are polarized and it is intended for GOSAT-2 to contain polarized channels as well, but for this study we will assume that the 'P' and 'S' components have been combined to form a non-polarized spectrum, since the primary aim of the polarization is to study atmospheric scattering and this is less important in this study, especially in the application of the proxy method.

We assume in this study that measurements are only made over land surfaces, thus other state vector elements required to make retrievals over water surfaces (such as wind speed) are not considered. We choose to ignore the specular reflectance effects of the sea since this requires a far more complex model than the Lambertian model employed by Yoshida et al. (2011) and thus will add additional computation time. Logically if there is not enough information present in high albedo land surface conditions such as deserts, then water glint reflectance is unlikely to have a positive impact.

In the TIR waveband, physical effects such as surface emissivity and total column temperature variations can have significant effects on retrievals, however, this work purely focuses on the SWIR, thus such elements are not considered in this study.

Clouds and aerosols can severally impact accurate measurements of methane; for the purposes of this study, we assume that all retrievals are from clear sky and unaffected by clouds and that any impact from aerosols is largely negated by the aforementioned proxy effect, thereby negating any need to account for aerosols in the light path.





## 5. Sensitivities of Bands 2 and 3

Given the sparseness of methane isotopologue lines, it is important to initially consider the relative sensitivity of the isotopologue lines in comparison to the interfering gases in the same spectral regions in both bands 2 and 3. These sensitivities are calculated from the Jacobean elements for each layer of the atmosphere using the ORFM tool and the

HITRAN2012 database as a basis for these calculations.

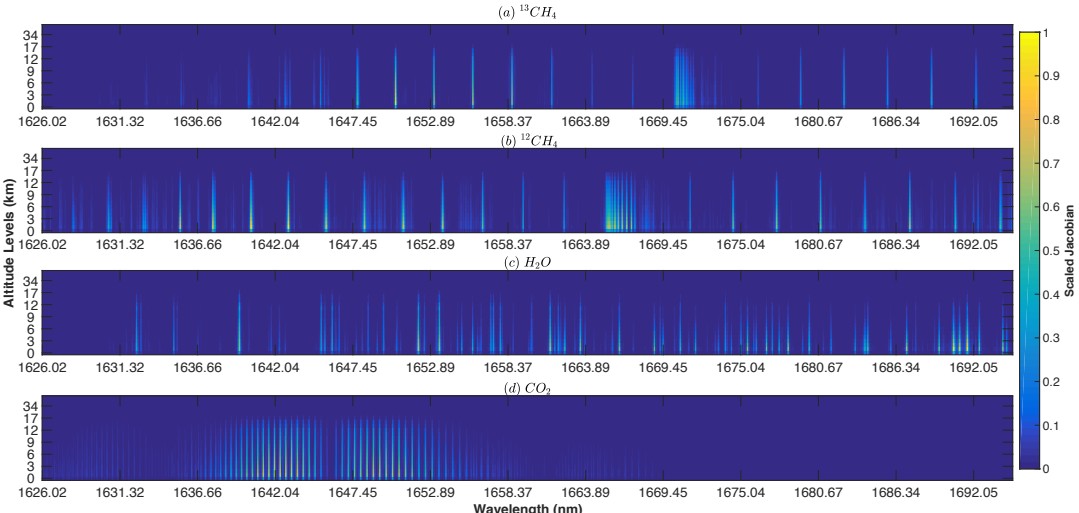

**Figure 2. Normalised sensitivity (between 0 and 1) of GOSAT-2 measured radiance with respect to a variation of 1% of the concentration of the main constituent gases in the 1626-1692 nm (band 2) wavelength range ($^{13}CH_4$ (a), $^{12}CH_4$ (b), $H_2O$ (c) and $CO_2$ (d)). All calculations were performed using the ORFM RTM, assuming a solar inclination angle of 30°, a satellite inclination angle**

**of 0° and a surface albedo of 0.1, in conjunction with the HITRAN2012 database and the University of Leicester model MIPAS atmosphere. All simulations were run at a 0.01 cm$^{-1}$ spectral resolution, and then convolved with a GOSAT-TANSO-FTS ILSF downloaded from the GOSAT Data Archive Service (https://data2.gosat.nies.go.jp/).**

Initial consideration is given to band 2, where solar irradiance is at a maximum for TANSO-FTS-2. Figure 2 shows the scale of the task at hand, with very few spectral lines of $^{13}CH_4$ present in this particular waveband, with only a handful indicating

significant sensitivity. We note that the $^{13}CH_4$ spectral lines exhibit similar behaviour to the $^{12}CH_4$ spectral lines, but are phase shifted by several nanometres, which is a characteristic of similar rotation-vibrational bands at different transition energies. This spectral region is known as the Tetradecad region and $^{13}CH_4$ absorption is dominated by the $2\nu_3$ vibrational band, and $^{12}CH_4$ in this region is characterised by more complex ro-vibrational states, which are described in more detail in Brown et al. (2013) and Lyulin et al. (2010). It is important to note that Brown et al. (2013) issue warnings that significant

uncertainties are still attached to the quantum positions of the methane isotopologue lines, especially relating to the effects of atmospheric broadening. This is less applicable to the wave range shown in Fig. 2, but should still be considered, suggesting





that uncertainty values must be ascribed to the centre position of the isotopologue lines. The methane lines in this region were significantly updated from the previous iteration in HITRAN2008 (Rothman et al., 2009), and were specifically recorded using pure $^{13}CH_4$ and differential absorption spectroscopy (Lyulin et al., 2010). They are judged to be accurate, however, Brown et al. (2013) note that "the new HITRAN list of $^{13}CH_4$ above 6170 cm$^{-1}$ is believed to be incomplete",

suggesting that there are additional lines in band 2 that could be leveraged in future HITRAN iterations. Figure 2 identifies that both $^{12}CH_4$ and $^{13}CH_4$ radiance sensitivity peaks at roughly the same altitudes (about 3 km), and remain sensitive up until the mid-troposphere. These results suggest that the transitions of the isotopologues are at similar lower-state energy levels and are therefore affected by atmospheric phenomenon such as temperature changes in similar fashions, contrasting with the sensitivities of carbon dioxide isotopologues as evidenced by Reuter et al. (2012). The MIPAS model atmosphere shows

water vapour concentrations dropping off very quickly with increasing altitude, and the sensitivity of water vapour in Fig. 2 shows significant variation in the altitudes at which water vapour is sensitive, but it also shows significant water vapour spectral lines in this spectral range suggesting that significant interference errors due to water vapour in the lower portion of the atmosphere can be expected. Carbon dioxide also has a significant presence in this waveband showing similar sensitivities to the main methane isotopologues.

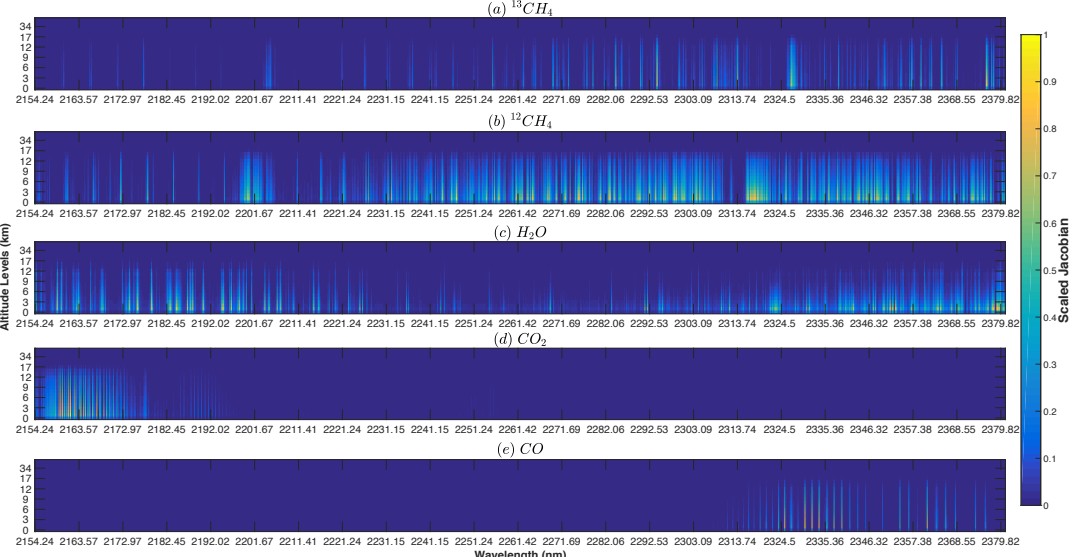

**Figure 3. Normalised sensitivity (between 0 and 1) of GOSAT-2 measured radiance with respect to a variation of 1% of the concentration of the main constituent gases in the 2154-2380 nm (band 3) wavelength range ($^{13}CH_4$ (a), $^{12}CH_4$ (b), $H_2O$ (c), $CO_2$ (d) and CO (e)). All calculations were performed using the ORFM RTM, assuming a solar inclination angle of 30°, a satellite**
**inclination angle of 0° and a surface albedo of 0.1, in conjunction with the HITRAN2012 database and the University of Leicester**





**model MIPAS atmosphere. All simulations were run at a 0.01 cm⁻¹ spectral resolution, and then convolved with a GOSAT-TANSO-FTS ILSF downloaded from the GOSAT Data Archive Service (https://data2.gosat.nies.go.jp/).**

Consideration is now given to a portion of TANSO-FTS-2 band 3 waveband, specifically the portion where methane spectral lines are particularly common (Fig. 3). Band 3 clearly has significantly higher levels of spectral lines for methane than band

2, although a wider waveband is considered since methane is present in only a very narrow spectral region of band 2. $^{12}CH_4$ is particularly prevalent in this region, showing high levels of sensitivity especially in the lower troposphere and in the boundary layer, while $^{13}CH_4$ lines are relatively dense and generally show low sensitivity except for a handful of lines. Sensitivity to the surface layer is well documented (Yoshida et al., 2011; Herbin et al., 2013), and like band 2, band 3 should be able to maximise measurements in the surface level as opposed to higher up in the atmosphere, where TIR measurements

tend to be more sensitive. We note that band 3 has lower spectral irradiance values than band 2 due to blackbody solar emissions, therefore lower radiance values from this region and lower SNR values are likely. This suggests that bands 2 and 3 have a trade off between the number of spectral lines present in the range, and the total SNR achievable by each band.

The ro-vibrational states in this particular spectral region (or polyad) are defined as the 'Octad', meaning that all $^{13}CH_4$ transitions exist at lower energy levels than the Tetradecad polyad of band 2 and that there are significantly lower numbers

of transitions available in band 3 as opposed to band 2. The spectral lines for $^{13}CH_4$ in band 3 are brand new in the HITRAN2012 database. All spectral lines were captured using FTIR measurements from the Kitt Peak facility and can be ascribed a high degree of confidence (Brown et al., 2013; Lyulin et al., 2010).

## 6. Establishing Information Content

Based on the equations and methods outlined in Sect. 4, the primary aim of this work is to determine the potential

information content of $^{13}CH_4$ in GOSAT-2/TANSO-FTS-2. However, unlike other GOSAT information content studies such as Herbin et al. (2013), there are no previous studies indicating the ideal retrieval set-up (i.e. surface or solar conditions, 'a priori' state vectors, etc.), we, therefore, are required to experiment in order to determine under what conditions there may be sufficient information available in the $^{13}CH_4$ bands to allow for an effective retrieval. To determine the potential information content in the bands, the following scenarios were designed, with the specific goal of varying the 'a priori' covariance

matrix, with the scaling factor included in all scenarios and the solar inclination angle, to determine what effects varying the optical path length may have:



**Table 3. Description of scenarios undertaken to determine potential $^{13}CH_4$ content in GOSAT-2/TANSO-FTS-2**

|  | BAND | COVARIANCE MATRIX | SOLAR INCLINATION ANGLE (°) | FIGURE RESULTS REPRESENTED IN |
|---|---|---|---|---|
| **SCENARIO 1** | 2 | Pure Diagonal | 30 | Fig. 4(a) & Fig. 5(a) |
| **SCENARIO 2** | 2 | Pure Diagonal | 60 | Fig. A1(a) & Fig. A3(a) |
| **SCENARIO 3** | 2 | Non-Diagonal | 60 | Fig. A2(a) & Fig. A4(a) |
| **SCENARIO 4** | 3 | Pure Diagonal | 30 | Fig. 4(b) & Fig. 5(b) |
| **SCENARIO 5** | 3 | Pure Diagonal | 60 | Fig. A1(b) & Fig. A3(b) |
| **SCENARIO 6** | 3 | Non-Diagonal | 60 | Fig. A2(b) & Fig. A4(b) |
| **SCENARIO 7** | 2 & 3 | Pure Diagonal | 30 | Fig. 4(c) & Fig. 5(c) |
| **SCENARIO 8** | 2 & 3 | Pure Diagonal | 60 | Fig. A1(c) & Fig. A3(c) |
| **SCENARIO 9** | 2 & 3 | Non-Diagonal | 60 | Fig. A2(c) & Fig. A4(c) |

The scenarios listed in Table 3 aim to determine the level of information content available in each of the SWIR bands, and a
combination of the bands. In addition to solar inclination angle, surface albedo is taken into account in each of the scenarios,
assuming a range of 0.1-0.6, consistent with vegetation to desert surface conditions. We note that Yoshida et al. (2011)
retrieve $CH_4$, $CO_2$ and $H_2O$ simultaneously; all gases are simulated to be retrieved simultaneously in this study as well. It is
necessary to retrieve $^{12}CH_4$ and $^{13}CH_4$ simultaneously in order to define a $\delta^{13}C$ value for any given retrieval. As identified in
Sect. 4, the TANSO-FTS-2 SNR is modified in accordance with the surface type. All other information is constant as either
specified in Sect. 4, or in the MIPAS model atmosphere (such as other gases or temperature profiles). Clear-sky conditions
are assumed (i.e. no clouds or aerosols), and no modifications of the optical path are expected.

Results from scenarios 1, 4 and 7 are shown in Fig 4. below. The results from scenarios 2, 5 and 8 are shown in Fig A1. and
scenarios 3, 6 and 9 are shown in Fig. A2 both of which are in the Appendix.

**6.1. Band 2**

Figure 4(a) shows the DOFS for $^{13}CH_4$ assuming retrievals from band 2 of TANSO-FTS-2, assuming the conditions outlined
in scenario 1 and the 'a priori' covariance variability identified in Sect. 4. For an $f$ factor of 1, equating to a $(10\%)^2$
variability in the covariance matrix, Fig. 4(a) suggests that an average of 0.1 DOFS can be expected for surface albedo
conditions varying between 0.1 and 0.6, suggesting any information for such a covariance matrix is strongly dependent on
the 'a priori' rather than the measurement. This is likely to be a product of the low concentrations of $^{13}CH_4$ in the





atmosphere. Figure 4(a) suggests that retrievals in band 2 for $^{13}CH_4$ is difficult, with only high albedo surface conditions giving the potential for unity values of DOFS, and even this only occurs when the $f$ factor is equal to 10 or over, equating to $(100\%)^2$ variability in the covariance matrix. As a comparison, Yoshida et al. (2011) show that DOFS of up to 2 for high albedo conditions are achievable for methane retrieval, with a covariance matrix roughly equivalent to $(\sim10\%)^2$ variability,

using TANSO-FTS. These results add to the weight of evidence that implies the difficulty of operational retrieval of $^{13}CH_4$. The maximum DOFS obtainable with band 2 assuming scaling factors up to the value 10, with the scenarios outlined in Table 3 are summarised in Table 4 below, and the related Figs are shown in the appendices below (Figs. A1(a) & A2(a)).

The results shown in Table 4 suggest that solar inclination angle is not an important factor in retrieval for band 2. There may

be some benefit to extreme solar inclination angle (e.g. sunglint or hotspot), and large satellite inclination angles but if significant information can only be obtained at extreme angles, this instantly removes the vast majority of GOSAT-2 measurements as beneficial. The inclusion of "off diagonal" values into the covariance matrix improves the information content of the signals, and in the case of a high scale factor, DOFS values of unity are obtained for all surface albedo values (see Fig. A2(a)), with unity being achieved for a scale factor of 5 $((50\%)^2$ variance) for an albedo value of 0.6. Even with this

best case scenario, as a single measurement, this is not significantly beneficial, and will not allow for an accurate value of $\delta^{13}C$ to be calculated nor allow for conclusions to be drawn about the nature of the source of the measurement. Overall, measurement variance can be decreased by averaging many measurements over large spatial regions or temporal periods, at the cost of high seasonal or spatial resolution.

**6.2. Band 3**

Figure 4(b) shows the DOFS for $^{13}CH_4$ assuming retrievals from band 3 of TANSO-FTS-2, assuming the conditions outlined in scenario 4 and the 'a priori' covariance variability identified in Sect. 4. Figure 4(b) suggests that an average of 0.3 DOFS can be expected for surface albedo conditions varying between 0.1 and 0.6 when assuming a variance of $(10\%)^2$, this is an improvement on the DOFS suggested by band 2, but not by a significant amount. Again this suggests that any information

for such a covariance matrix is strongly dependent on the 'a priori' rather than the measurement. However, we note that unlike the results shown in Fig. 4(a), DOFS values of 1 can be expected above a scale factor of 7 for all albedos shown; suggesting that definitive information from band 3 for $^{13}CH_4$ can be expected, and is not reliant on extremely high surface albedo conditions, which will be rare on the surface of the Earth at these wavelengths. Yet the required scale factor is still high and given this level of variance, significant spatial and temporal averaging are most likely required. The results from

the remaining band 3 scenarios are shown in Table 4 and Figs. A1(b) and A2(b) below.



Like band 2, changing the solar inclination angle does not have a significant impact on the DOFS available for $^{13}CH_4$ retrieval. However, the addition of "off diagonal" elements to the 'a priori' covariance matrix has a significant impact on the available DOFS as highlighted by Fig. A2(b), which suggests that information can be extracted from a total column for all surface albedos at a scaling factor of 4 $(40\%)^2$, increasing to a factor of 2.5 $(25\%)^2$ when only considering high surface

albedo values. It is clear that band 3 of TANSO-FTS-2 has significant benefits over band 2 in terms of information content, even without exactly fixed instrument noise or 'a priori' state vectors and covariance matrices, there is a clear benefit in retrieving $^{13}CH_4$ in this band. However, the results suggest that significant variance is still required in order to guarantee the solution to the OEM is based on the measurements rather than the 'a priori', therefore substantial temporal and spatial averaging is likely to be required in the same manner as band 2.

**6.3. Combined Band 2 and Band 3**

The dual detector nature of the future SWIR bands of GOSAT-2 allows for a combination of the information channels of bands 2 and 3 in order to maximise information content. Because both bands are based on solar backscatter measurements, and largely contain the same interfering elements, there is no issue with a direct combination. On the other hand, including TIR elements which are sensitive to different portions of the atmosphere, are difficult to combine directly with SWIR

measurements (Herbin et al., 2013). The pure application of this concept increases calculation time significantly due to the number of spectral lines present in both bands; however, full retrievals are not the main aim of this work as opposed to determining maximum information content in the GOSAT-2 bands, therefore we are justified again in making retrieval speed a low priority.

The results identified in Fig. 4(c) show some differences between the application of spectral lines in band 3 and those in band 2 & band 3. Although these differences are minor, there is a definite increase in DOFS of roughly 0.1 for each of the scaling factors. We also note that the spread of the DOFS lines due to varying albedo conditions are more widely spaced as compared to band 3 DOFS, suggesting that measurements in the combined bands are more sensitive to surface conditions. The degree to which the DOFS increases w.r.t. the scaling factor is sharper than in band 2 so that DOFS values of unity are

achieved for all albedo values at scaling factor 7, a variance of $(70\%)^2$. These values suggest that retrieval of $^{13}CH_4$ is feasible within the operational lifetime of GOSAT-2. This is further emphasised by the results from the other bands 2 & 3, summarised in Table 4 below, and Figs. A1(c) & A2(c).

Table 4 shows the same trends as the DOFS results shown in bands 2 and 3 individually, in that the solar inclination angle

has a minor impact on the DOFS, and the variation of the 'a priori' covariance matrix has a similar scaling effect on the DOFS. Yet the combination of both bands has yielded a modest increase in the DOFS for all scenarios at all surface albedos. Considering the 'off diagonal' 'a priori' case in scenario 9 we find that DOFS equal to unity are achievable for all surface albedo type at a scaling factor of 3.5 $(35\%)^2$, which is clearly superior to any of the other cases considered in this manuscript.



Therefore there are significant benefits to dual band retrievals with TANSO-FTS-2. However, it is important to note that combining the two bands led to a significant computation cost, and is possibly not practical for full scale retrievals in the form identified in this manuscript.

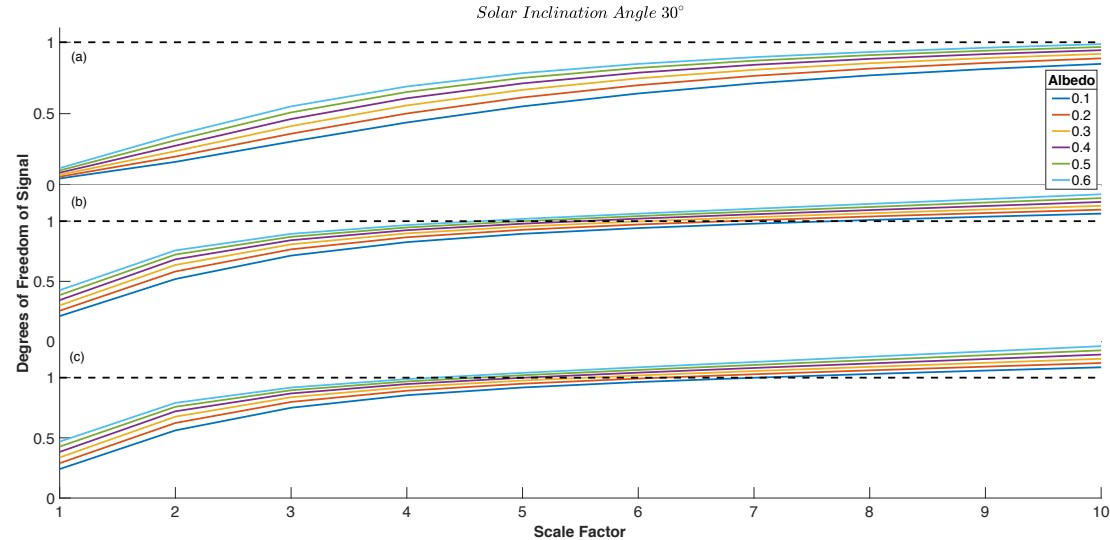

**Figure 4. Degrees of Freedom for $^{13}CH_4$ vs scaling factor '$f$' for scenarios outlined in Table 3. Each coloured line represents different surface albedo conditions as shown in the key. The black dashed line represents unity DOFS: a) indicates scenario 1, b) indicates scenario 4, c) indicates scenario 7.**

All the maximum achievable DOFS results from all scenarios are outlined in Table 4 below.

**Table 4. Summarisation of DOFS characteristics for all scenarios 6 surface albedo levels.**

### Max DOFS for Surface Albedo

| Albedo | 0.1 | 0.2 | 0.3 | 0.4 | 0.5 | 0.6 |
|---|---|---|---|---|---|---|
| Scenario 1 DOFS | 0.85 | 0.89 | 0.92 | 0.94 | 0.97 | 0.99 |
| Scenario 2 DOFS | 0.85 | 0.89 | 0.92 | 0.94 | 0.97 | 0.99 |
| Scenario 3 DOFS | 1.03 | 1.06 | 1.08 | 1.10 | 1.13 | 1.15 |




| | | | | | | |
|---|---|---|---|---|---|---|
| *Scenario 4 DOFS* | 1.06 | 1.09 | 1.13 | 1.16 | 1.19 | 1.22 |
| *Scenario 5 DOFS* | 1.06 | 1.09 | 1.12 | 1.15 | 1.18 | 1.22 |
| *Scenario 6 DOFS* | 1.26 | 1.31 | 1.36 | 1.41 | 1.46 | 1.50 |
| *Scenario 7 DOFS* | 1.08 | 1.12 | 1.16 | 1.19 | 1.23 | 1.26 |
| *Scenario 8 DOFS* | 1.08 | 1.12 | 1.15 | 1.19 | 1.22 | 1.25 |
| *Scenario 9 DOFS* | 1.30 | 1.35 | 1.41 | 1.46 | 1.51 | 1.55 |

**7. Error Analysis**

Even if sufficient DOFS can be established to identify where $^{13}CH_4$ retrievals are influenced more by the measurement than by the 'a priori', the errors associated with the retrieval may well make identifying methane source types a practical impossibility. Therefore an assessment of the expected total column errors is required, these errors for $^{13}CH_4$ can be summarised as (Yoshida et al., 2011):

$$\sigma = \frac{\sqrt{h^T S h}}{h^T 1} \quad (16)$$

$$h^T = \left( w_{dry,1} w_{dry,2} \dots w_{dry,n} \right) \quad (17)$$

where $\sigma$ is the total column 'a posteriori' error, depending on the subset of altitudes or pressures used, $h$ is the dry air partial column, calculated from $n$ layers of the retrieval grid (21 in this case) where $w_{dry}$ is calculated from the model pressure profile and $H_2O$ concentration profile. $1$ is a column vector with elements of unity and a length equivalent to the dry air partial column.

Using the scenarios outlined in Table 3, the total column error (along with the interference, smoothing and measurement errors) for $^{13}CH_4$ retrieval can be established. Total column errors for $^{12}CH_4$ are assumed to be documented in studies such as Parker et al. (2011) and Yoshida et al. (2011).

Results from scenarios 1, 4 and 7 are shown in Fig 5. Below. The results from scenarios 2, 5 and 8 are shown in Fig A3. and scenarios 3, 6 and 9 are shown in Fig. A4 both of which are in the Appendix.





### 7.1. Band 2

Based on the summation of errors identified above, we can estimate the precision of a synthetic retrieval in band 2 of TANSO-FTS-2, for the range of 'a priori' covariance matrices identified previously.

Scenario 1 (Fig. 4(a)) suggests that retrievals are heavily biased towards the 'a priori' in band 2 of TANSO-FTS, except perhaps for a variance of $(100\%)^2$, over a very bright surface (i.e. albedo of 0.6, or SNR equal to 500). In this case, the maximum precision for a single sounding equates to 2.4ppbv for $^{13}CH_4$. Based on the total column averaged concentration of $^{13}CH_4$ from the MIPAS profile identified in Sect. 4.1, 2.4ppbv precision equates to roughly 13% error. For reference, Yoshida et al. (2011) show that the average total column precision for methane retrievals is 5.86ppbv, which equates to

3.4%. Based on the DOFS values for scenario 2 (Fig. A1(a)), we can assume similar precision values, however, the DOFS values for scenario 3 (Fig. A2(a)) suggest that unity is achieved for variance of $(80\%)^2$ or above for the whole SNR range. Based on the error values shown in Fig. A4(a), the maximum precision at this variance is 2.2ppbv, and the minimum is 3ppbv, equating to 11.5% error and 15.6% respectively. These results are far from the 0.25ppbv precision requirement set in the introduction, however, the precision can be increased through averaging multiple retrievals together (both temporally and

spatially), where the standard deviation is inversely proportional to the square root of the number of measurements. Therefore for scenario 3, we suggest that a precision of 0.25ppbv can be achieved using the average of 78 measurements for high SNR, and 144 measurements for low SNR, both of which are achievable over a significant period of time with a large spatial region for GOSAT-2.

### 7.2. Band 3

Using the methods identified above in band 2, the total errors for band 3 retrievals are explored.
Scenario 4 (Fig. 4(b)) shows that unity DOFS occurs for variance of $(80\%)^2$ or above for the whole SNR range. Using Fig. 5(b), we can suggest that the maximum and minimum precision is 1.5ppbv and 1.8ppbv respectively, these equate to 7.8% and 9.4% of the total column. We suggest that through spatial and temporal averaging, these errors can be reduced to the

0.25ppbv target by averaging 36 and 52 measurements for maximum and minimum SNR cases. Changing the solar inclination angle for scenario 5 (Fig. A1(b)) does not impact the DOFS significant, however, if we consider scenario 6 (Fig. A2(b)), DOFS of unity occur for variance of $(40\%)^2$ or above for the whole SNR range. The maximum and minimum precisions at this variance are 1.1ppbv and 1.3ppbv (Fig. A4(b)) respectively. The target precision can be increased through averaging 20 and 28 measurements respectively. Therefore for retrievals with band 3, we suggest that measurements of $\delta^{13}C$

to an accuracy of 10 per mil can be achieved within monthly periods of GOSAT-2 measurements.





### 7.3. Combined Band 2 and Band 3

Scenario 7 (Fig. 4(c)) shows that unity DOFS occurs for variance of $(70\%)^2$ or above for the whole SNR range. Using Fig. 5(c), we suggest that the maximum and minimum precision at this variance is 1.5ppbv and 1.8ppbv respectively, i.e. very similar to those found in band 3 scenario 4, therefore similar numbers of measurements are required in order to achieve the desired precision. If we consider scenario 9, unity DOFS are achieved for a variance of $(35\%)^2$ (Fig. A2(c)), the maximum and minimum precisions at this variance are 0.7ppbv and 1.2ppbv (Fig. A4(c)) respectively. The target precision can be increased through averaging 8 and 24 measurements respectively. Scenario 9 shows the best results in terms of information content, and measurement precision, to the point where over highly reflective surfaces very few measurements are required in order to make an accurate assessment of the source type.

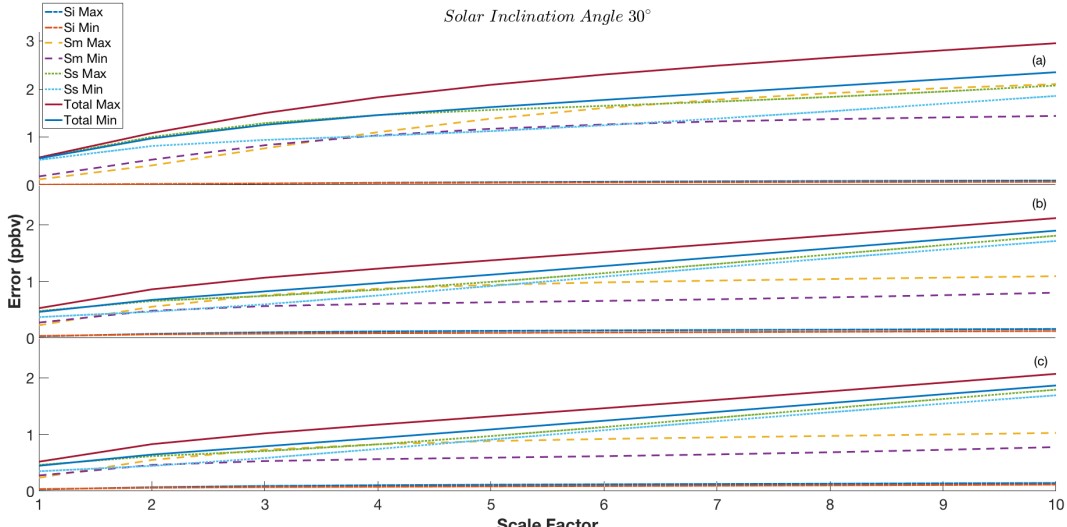

**Figure 5. Synthetic total column retrieval errors retrieval for scenarios outlined in Table 3, based on Eqs. (8,9,10 & 16). The maximum and minimum values (based on maximum and minimum SNR), for interference, measurement and smoothing error are shown, in addition to the total error: a) indicates scenario 1, b) indicates scenario 4, c) indicates scenario 7.**





### 8. Conclusions and Summary

To summarise, this work investigates the possibility of whether $^{13}CH_4$ can be retrieved with a sufficient level of accuracy by bands 2 and 3 of the GOSAT-2/TANSO-FTS-2 instrument, in order to make a judgement on the nature of a methane source type (biogenic, thermogenic or abiogenic), via the use of the $\delta^{13}C$ ratio. We assume that an accuracy of 10 per mil of $\delta^{13}C$

values is sufficient to distinguish between methane source types, as shown by Rigby et al. (2012), and with this accuracy, we calculate that a $^{13}CH_4$ retrieval precision of 0.25ppbv is required in order to achieve $\delta^{13}C$ with a 10 per mil accuracy.

Using the well established DOFS methods (Rodgers, 2000) the RTM ORFM, and the assumption of clear sky conditions we calculate the key metrics of DOFS and total retrieval error in order to judge a) the information content in a retrieval, and b) the precision of that retrieval, based on a series of test 'a priori' covariance matrices. Using a combination of bands 2 and 3,

we find that total column retrieval of $^{13}CH_4$ with sufficient DOFS is possible, with a maximum and minimum precision of 0.7ppbv and 1.2ppbv respectively. Assuming statistical error reduction techniques, this precision can be increased to 0.25ppbv by averaging over 8 and 24 measurements respectively. This number of measurements is certainly achievable over a monthly period, assuming modest spatial sampling of 2°x2°, which is often how GOSAT data is represented. Implying that GOSAT-2 will be able to differentiate between methane source types at a high temporal resolution.

This analysis was also applied to bands 2 and 3 individually, and it was found that band 2 can achieve enough DOFS for a $^{13}CH_4$ retrieval at the desired precision, based on averaging up to 144 measurements for a completely unconstrained 'a priori' covariance matrix. Band 3 showed similar results to bands 2 and 3 combined but required up to 28 measurements in order to achieve the required precision.

Across all bands, we find that the DOFS and precision are significantly affected by the instrument SNR, but not by the solar

inclination angle to any significant degree. In addition, the rate of increase of DOFS with respect to scaling factor is significantly higher in the combined bands, than either band when considered individually. However, combining the DOFS from both bands leads to a significant computational penalty.

Given the relative abundance of $^{13}CH_4$ spectral lines in band 3 of GOSAT-2, there is also some scope for future comparisons with measurements from Sentinel 5/5-P, both of which are sensitive to the same spectral regions. In addition, it is envisaged

that there will be a period where GOSAT and GOSAT-2 are in simultaneous operation, and it may be possible to combine the measurements from both of these satellites in order to reduce uncertainty in isotopologue measurements.

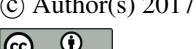


## 9 Appendices

The scenario plots not indicated in the main text are shown below, namely scenarios 2, 3, 5, 6, 8 and 9 for both DOFS and retrieval errors.

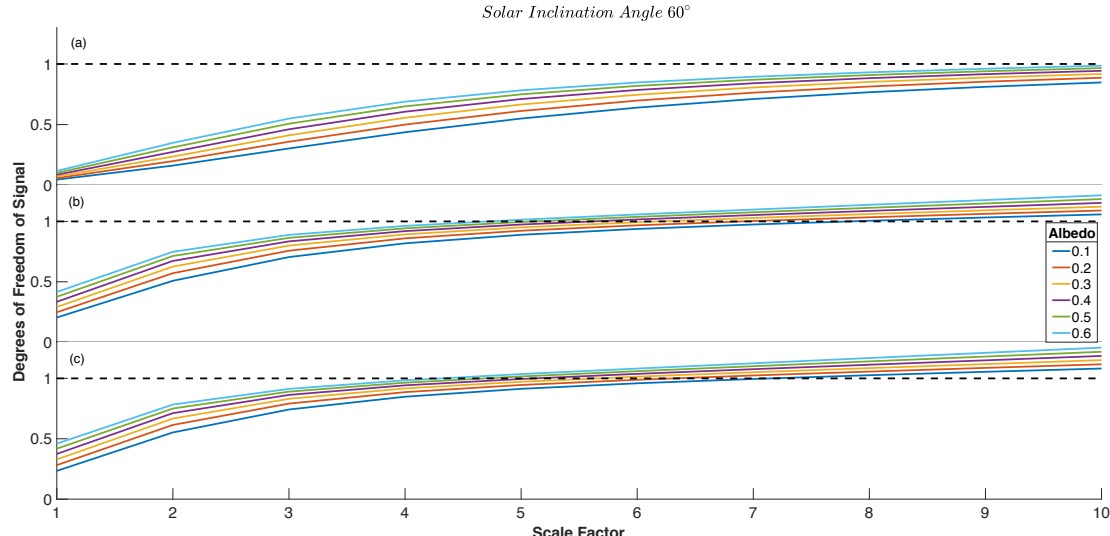

**Figure A1. Degrees of Freedom for $^{13}CH_4$ vs scaling factor 'f' for scenarios outlined in Table 3. Each coloured line represents different surface albedo conditions as shown in the key. The black dashed line represents unity DOFS: (a) indicates scenario 2, (b) indicates scenario 5, (c) indicates scenario 8.**





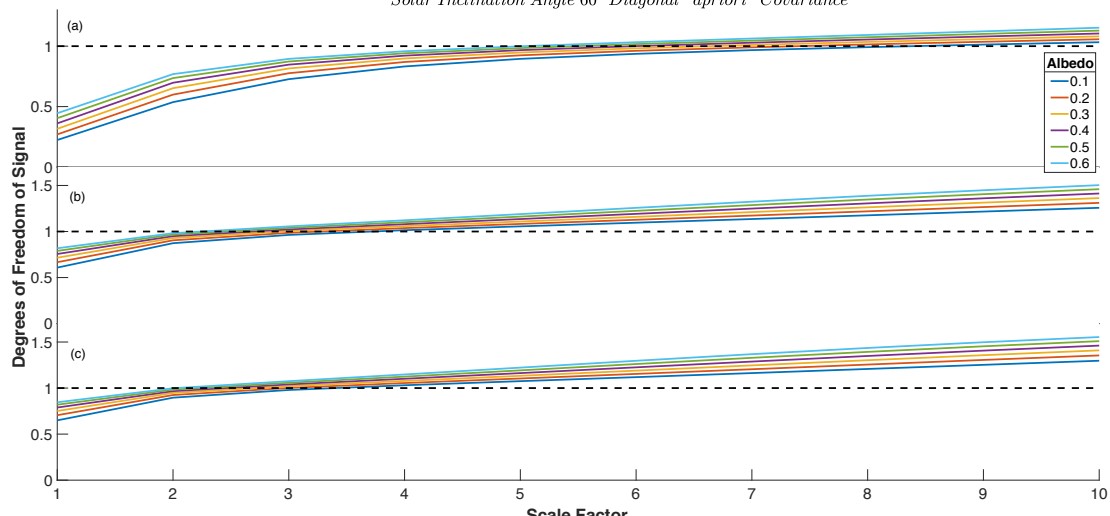

**Figure A2. Degrees of Freedom for $^{13}CH_4$ vs scaling factor 'f' for scenarios outlined in Table 3. Each coloured line represents different surface albedo conditions as shown in the key. The black dashed line represents unity DOFS: (a) indicates scenario 3, (b) indicates scenario 6, (c) indicates scenario 9.**

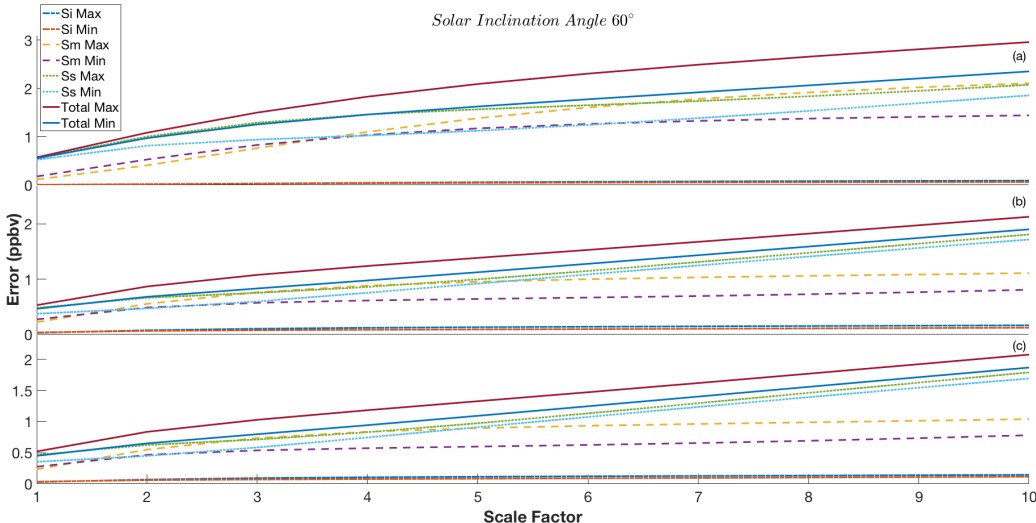





**Figure A3.** Synthetic total column retrieval errors retrieval for scenarios outlined in Table 3, based on Eqs. (8,9,10 & 16). The maximum and minimum values (based on maximum and minimum SNR), for interference, measurement and smoothing error are shown, in addition to the total error: a) indicates scenario 2, b) indicates scenario 5, c) indicates scenario 8.

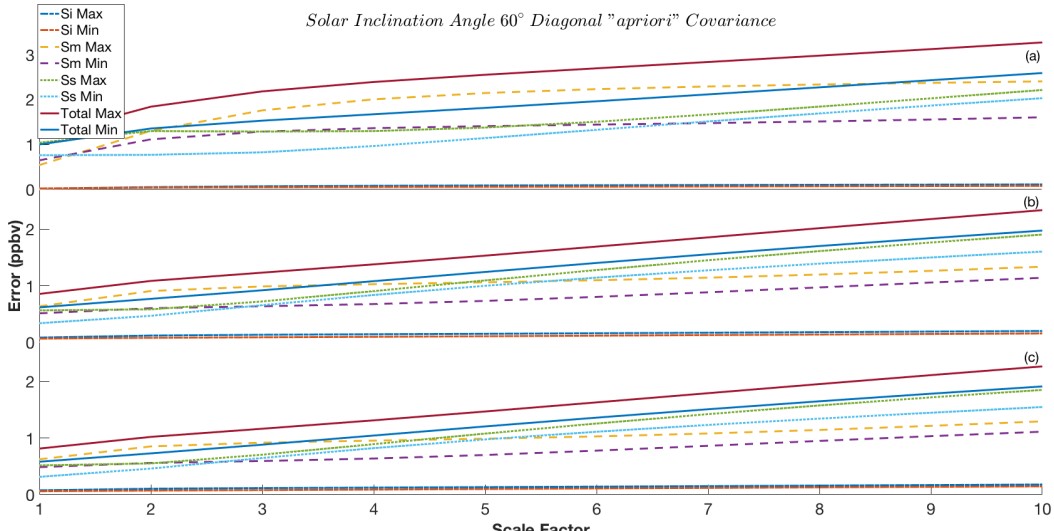

**Figure A4.** Synthetic total column retrieval errors retrieval for scenarios outlined in Table 3, based on Eqs. (8,9,10 & 16). The maximum and minimum values (based on maximum and minimum SNR), for interference, measurement and smoothing error are shown, in addition to the total error: a) indicates scenario 3, b) indicates scenario 6, c) indicates scenario 9.

### Author Contribution

E.M., Y.Y., M.T. and J.-P.M. conceived and designed the experiments; E.M. performed the experiments, analysed the data and wrote the paper.

### Acknowledgements

15   This research was partially funded by NIES GOSAT-2 Project, in combination with the first author's PhD grant from the National Centre for Earth Observation (NCEO) through the Natural Environment Research Council (NERC) based in the UK (award number 157550). We would also like to acknowledge Anu Dudhia at Oxford University for the ORFM, and JAXA/NIES/MOE for GOSAT-TANSO-FTS data.



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
