# Peer review of "Information Content Analysis: The Potential for Methane Isotopologue Retrieval from GOSAT-2"

_Atmospheric Measurement Techniques, 2017_

## Referee Comment (RC1) · H. Hu (Referee) · 7 Sep 2017

H. Hu (Referee)

h.hu@sron.nl

Received and published: 7 September 2017

**General comments:**

This paper explores the possibility to retrieve the ratio between the two main methane isotopologues from a future GOSAT-2 instrument. In itself, this is a very interesting study, because, if possible, satellite data could be used in the future to discriminate between natural and anthropogenic sources of methane on a global scale. However, the results and conclusions of this study depend heavily on many assumptions, some of which are not quantitatively investigated.

Main points:

- Impact of precision of current methane retrievals, authors assume here 6ppbv from

Yoshida et al. (2011) which is an average and does not include forward model or instrument errors. However, an estimate of the methane precision from TCCON validation is around 15 ppbv (see e.g. Schepers et al. 2012 and Parker et al. 2015). This impact should be quantified or at least discussed.

- The authors experiment with different a priori covariance matrices, because no suitable one is known. It seems in this case Philips-Tikhonov regularization is more suitable than optimal estimation. Could the authors comment on that?

- The error analysis could be extended by perturbing the a priori  ${}^{13}CH_4$  profile with the assumed a priori errors and comparing the retrieved  ${}^{13}CH_4$  against the truth. Would this be the same as the derived precisions?

**Specific comments:**

- Abstract, page 1, line 22: Rephrase the following sentence for clarity: "Large unconstrained 'a priori' covariance matrices are required ... retrieval errors." Suggestion: "We find that large unconstrained covariance matrices are required in order to achieve sufficient information content, while the solar inclination angle has limited impact on the information content." The authors should avoid "retrieval errors" in this sentence, because that could suggest the solar zenith angle does not have an impact on the forward model retrieval error which is certainly incorrect.

- Introduction, page 2, line 19: This sentence seems incorrect: "Plant based photosynthesis enzymes discriminate against carbon dioxide during uptake..." Should it be: "Plant based photosynthesis enzymes discriminate against 13C during carbon dioxide uptake..."?

- In general, the introduction should not contain formulas and derivations. I suggest to move Eq. (1) to a subsection where the requirements on the errors are derived. At the same time, the derivation of the minimum precision of 0.25 ppbv on the  $^{13}\mathrm{CH}_4$  retrievals should be made explicitly (e.g. assumed values, error propagation), since

AMTD
this is of such importance to the rest of the paper. In particular, the assumption of a precision of 6ppbv for methane retrievals from Yoshida et al. (2011) is too optimistic, since that is an average and errors up to 15 ppbv can occur. Also, the systematic error is not included in his work or in Yoshida et al. (2011) as stated in the paper, at least a discussion about the impact of the systematic error on the conclusion/results should be given.

- Introduction, page 3, 7: Rephrase/shorten long sentence: "Some measurements from balloon soundings ...to the scientific community"

- page 4, section 2.1: Mention here that the RTM does not include scattering.

- page 8, section 4.1: First the authors attempt to determine the variance in  ${}^{13}CH_4$  by taking the maximum range of observed  $\delta^{13}C$ , this approach is a rough approximation at best. From that they derive  $(3\%)^2$ , but nevertheless take  $(10\%)^2 - (100\%)^2$  in their study. This seems random. Please reformulate or justify better why  $(10\%)^2 - (100\%)^2$  is reasonable. Also, it not explained in which cases a diagonal covariance matrix is more likely and in which cases an off-diagonal one.

- page 12/13, Figure 2 and 3: The color plots are not clear, please use other color scale or representation.

- page 17/18: It is mentioned that the combined band 2 and band 3 retrieval significantly increases computation time compared to band 2 or band 3 retrieval. Stating the CPU time for all cases would be useful to make that point.

- I am missing a discussion on how methane isotolopogue retrievals, if successful, could be validated. Please include a discussion on possible validation strategies, e.g. using NOAA measurements.

**Technical corrections:**

-"plant-based" or" plant based", use one consistently throughout the text

- page 7, line 13: definite -> define

---

## Referee Comment (RC2) · Anonymous Referee #2 · 10 Nov 2017

Methane isotopologue detection is important for methane source detection. Measurements of isotopologues from satellite would be very important. The paper by Malina et al. is an interesting and informative paper on methane isotopologue detection capability, and is very relevant for GOSAT-2 and other future SWIR spectrometers. In general the story of the paper is well-written (although the text somewhat sloppy) and the figures are clear. The paper is suited for AMT.

The paper can be accepted after the following comments are addressed.

Main comments:

- Sect. 2.1: A better description of the applicability of the limb sounding forward model ORFM for a nadir viewing instrument like GOSAT-2 is needed. There are missing

processes in the ORFM model, like atmospheric scattering. Is surface reflection well included? Is surface elevation included?

- The term "solar inclination angle" is not used in nadir remote sensing. Therefore, this term should be converted to the term "solar zenith angle", which is 90 degrees – solar inclination angle. Please use the symbol \theta_0 for the solar zenith angle and \theta for the viewing zenith angle.

- P. 8, l. 17: What is the basis of setting the scaling factor f to (10 %)2 variance ? This means that you needed a much larger deviating delta13C than can be anticipated. What is the basis of that assumption?

Minor and Textual comments:

- P. 2, l. 6: acronym GHG was already explained on the previous page

- P. 2, l. 28: please give a reference for VPDB

- P. 3, l. 18: 6ppbv: please add a space between the number and the unit. This holds throughout the paper, at many places, for many quantities, including %.

- Table 1: please give the spectral resolution of the bands.

- Eq. 7: DOFS: acronyms should not be in italics because they are not symbols

- P. 11, l. 4: please give an example of such errors.

- P. 11, l. 18: channel > channels

- P. 11, l. 30-32: this should be mentioned earlier.

- P. 12, l. 17: why a comma after 2\nu3?

- P. 13, l. 8: phenomenon > phenomena

- P. 14, l. 1-2: All simulations ..: please add this information to the main text because it is important information.

- P. 14, l. 10: spectral irradiance > solar irradiance

- At the same line: please remove: "due to blackbody solar emissions" (which is a strange comment)

- P. 14, l. 20-23: too long sentence - please make shorter sentences.

- P. 15, l. 6: At which wavelength do these albedo values hold? Please give a reference for these surface albedo values.

- P. 15, l. 17: please refer to Eq. 11 for the definition of f

- P. 16, l. 9: why does the inclination angle not matter? This is unexpected. The air mass is much larger at smaller inclination angles.

- P. 16, l. 10: The hotspot depends on the scattering angle, not on the inclination angle; the sun glint depends on viewing and solar geometry.

- P. 16, l. 11: extreme angles > special geometries

- P. 16, l. 23: " . . ., this is an . . .": please start a new sentence

- P. 17, l. 8: . . .therefore: please start a new sentence

- P. 17, l. 13: remove: including

- P. 17, l. 33: manuscript > paper (also on next page)

- Caption fig. 4: Degrees of Freedom for Signal

- Caption Table 4: Summarisation > Summary; and 6 surface albedos

- Table 4: remove DOFS in the left-hand column since it is superfluous

- P. 19, l. 20: remove the points around below.

- P. 20, l. 5: a priori should be in italics, and not in quotation marks (throughout the paper)

- Caption Fig. 5: remove the second word retrieval

- Caption Fig. A1: f should not be in quotation marks, since it is a normal symbol

Figures:

Fig. 1: please give the unit of the x-axis

References:

The references are very sloppy. First of all, the authors should replace the URLs by normal journal references. Please replace capital font for titles by normal font. Aydin et al.: the journal name is missing.

---

## Author Comment (AC1) · 20 Dec 2017

Dear Dr. Hu, thank you for your review of our manuscript, and the comments below. In order to respond to your comments, we have kept your original comments in black non-italics. Our responses are in bold blue italics, and changes to the manuscript are in bold blue underlined italics.

**General comments**: This paper explores the possibility to retrieve the ratio between the two main methane isotopologues from a future GOSAT-2 instrument. In itself, this is a very interesting study, because, if possible, satellite data could be used in the future to discriminate between natural and anthropogenic sources of methane on a global scale. However, the results and conclusions of this study depend heavily on many assumptions, some of which are not quantitatively investigated.

Thank you for raising these points, we address them below.

Main points:

- Impact of precision of current methane retrievals, authors assume here 6ppbv from Yoshida et al. (2011) which is an average and does not include forward model or instrument errors. However, an estimate of the methane precision from TCCON validation is around 15 ppbv (see e.g. Schepers et al. 2012 and Parker et al. 2015). This impact should be quantified or at least discussed.

Thank you for raising this point. We have now expanded a discussion on how the precision of the CH4 measurements will impact the required 13CH4 precision. The exact details of which are shown in the specific comments section below. However to summarise, the 13CH4 precision is calculated from the range of 13CH4 values that fall into a 10‰  $\delta^{13}$ C change, for a given 12CH4 value. The precision of 12CH4 is important as this limits the range of 13CH4 values available for the  $\delta^{13}$ C change.

In general however, we do not assume that precision errors will be as large as 15 ppbv, since both of these papers use spatial matchup criteria of +/-5 deg when compared against TCCON.

- The authors experiment with different a priori covariance matrices, because no suitable one is known. It seems in this case Philips-Tikhonov regularization is more suitable than optimal estimation. Could the authors comment on that? *Philips-Tikhonov regularization would certainty be suitable in the case of retrieval methane isotopologues. However in the case of GOSAT-2, it is expected that the numerous algorithms currently applied to GOSAT which use a priori covariance matrices (e.g. Parker et al., (2011;2015), Yoshida et al., (2011;2013)) will be applied to GOSAT-2 (via appropriate modification). This paper is aimed at those algorithms, and aims to provide the basic set-up required. We will consider a Philips-Tikhonov method for future follow up studies, and we have included the Philips-Tikhonov method as a discussion point in this manuscript.*

Please see section 9, Page 27, lines 5-13.

- The error analysis could be extended by perturbing the a priori 13CH4 profile with the assumed a priori errors and comparing the retrieved 13CH4 against the truth. Would this be the same as the derived precisions?

This would be an interesting activity, unfortunately at present the author does not have access to an iterative retrieval algorithm (only a linear information content analysis was applied), which we believe would be required in order perform the analysis that is suggested, otherwise we would be passing linear values backwards and forwards, and we are not sure if this would be beneficial to the study.

**Specific comments:**

- Abstract, page 1, line 22: Rephrase the following sentence for clarity: "Large unconstrained 'a priori' covariance matrices are required ... retrieval errors." Suggestion: "We find that large unconstrained covariance matrices are required in order to achieve sufficient information content, while the solar inclination angle has limited impact on the information content." The authors should avoid "retrieval errors" in this sentence, because that could suggest the solar zenith angle does not have an impact on the forward model retrieval error which is certainly incorrect. *Thank you, we have modified this sentence as requested. We have also changed solar inclination angle to solar zenith angle, as requested by reviewer 2.*

Changed, Page 1, lines 22-23.

- Introduction, page 2, line 19: This sentence seems incorrect: "Plant based photosynthesis enzymes discriminate against carbon dioxide during uptake..." Should it be: "Plant based photosynthesis enzymes discriminate against 13C during carbon dioxide uptake..."?

Thank you for spotting this. We have changed this sentence.

**Sentence changed, Page 2, line 19 to "discriminate against 13C carbon dioxide (13CO2)"**

- In general, the introduction should not contain formulas and derivations. I suggest to move Eq. (1) to a subsection where the requirements on the errors are derived. At the same time, the derivation of the minimum precision of 0.25 ppbv on the 13CH4 retrievals should be made explicitly (e.g. assumed values, error propagation), since this is of such importance to the rest of the paper. In particular, the assumption of a precision of 6ppbv for methane retrievals from Yoshida et al. (2011) is too optimistic, since that is an average and errors up to 15 ppbv can occur. Also, the systematic error is not included in his work or in Yoshida et al. (2011) as stated in the paper, at least a discussion about the impact of the systematic error on the conclusion/results should be given.

Thank you for these observations. We have moved Eq. (1) to a new subsection in section 2, including appropriate aspects of the introduction to give an

appropriate discussion to Eq. (1). This new subsection 2.1 also discusses how the required  ${}^{13}CH_4$  precision is derived. The precision is derived by determining what the change in  ${}^{13}CH_4$  concentration is for a 10 per mil change in  ${}^{13}C$ . This is calculated for a range of  ${}^{12}CH_4$  values (which are presented in 5 ppb steps), and is represented in the Figure below.

Figure 1. Range of expected terrestrial 13CH4 values (y-axis) given a range of 12CH4 values between 1770 and 1830 ppb, and  $\delta^{13}$ C between -80 and -10 per mil (x-axis). The diagonal solid lines represent the 12CH4 values for a given 12CH4 value, while varying the  $\delta^{13}$ C range. There are 13 12CH4 lines representing the 12CH4 range in 5 ppb steps. The red line (a) shows the 13CH4 change between -50 and -40  $\delta^{13}$ C for a 12CH4 of 1770 ppb; (b) is as (a), but includes a 12CH4 change of 5 ppb; (c) is as (a) and (b) but includes a 12CH4 change of 15 ppb.

Based on this figure, we agreed with your statement that we were too optimistic revised down the target 13CH4 precision to 0.2 ppb. This figure does not include precision errors on CH4. Using this figure, where each black diagonal line represents a CH4 step change of 5 ppb, we determined that a 5 ppb uncertainty in CH4 corresponds to a required 0.08 ppb increase in precision of 13CH4 (or 4 per mil  $\delta^{13}$ C), and a 15 ppb methane uncertainty corresponds to a required 0.16 ppb increase in precision of 13CH4 (or 8 per mil  $\delta^{13}$ C). We have updated the conclusions as necessary. We have also included a discussion on how an assumed 5 ppb bias on GOSAT methane measurements against TCCON corresponds to a  $\delta^{13}$ C bias of 4 per mil.

Changes to the introduction w.r.t. moving Eq.(1) to a new section are made on Page 2, lines 28-31; Page 3, lines 9-10, 15.

In regards to the changes described above, Section 2 has been renamed, Page 4, line 11, and the description of the section has been appropriately modified, Page 4, line 12. The new section 2.1, which includes Eq. (1), along with the description of the equation, and the derivation of target 13CH4 precision has been inserted at Page 4, lines 15-30, and Page 5, lines 1-20 and Page 6, lines 1-15.

Based on these changed, the original subsection 2.1 has been updated to subsection 2.2, Page 6, line 16, and the original subsection 2.2 has been updated to subsection 2.3, Page 7, line 13.

- Introduction, page 3, 7: Rephrase/shorten long sentence: "Some measurements from balloon soundings ...to the scientific community" *We have split this into two sentences.*

See Page 3, lines 9 and 10.

- page 4, section 2.1: Mention here that the RTM does not include scattering. *This has been included.*

**See Page 6, lines 32-33, and Page 7, lines 1-2.**

- page 8, section 4.1: First the authors attempt to determine the variance in 13CH4 by taking the maximum range of observed  $\delta$ 13C, this approach is a rough approximation at best. From that they derive (3%)2, but nevertheless take (10%)2-(100%)2 in their study. This seems random. Please reformulate or justify better why (10%)2-(100%)2 is reasonable. Also, it not explained in which cases a diagonal covariance matrix is more likely and in which cases an off-diagonal one.

Thank you for raising this point, which is an important point and raised by reviewer 2 as well. We accept that the  $(3 \%)^2$  figure is a rough approach, and discuss as such in the updated text.

We have now included a more in-depth discussion into why the  $(10\%)^2 \cdot (100\%)^2$ values are used. The reason for this is based on the relationship between the a priori covariance and the DOFS from the assumed GOSAT retrieval. From experience we know that methane covariance is often set to  $(10\%)^2$  variance, in order to allow for some variation in the retrieved solution. At this level of variance we can expect between 1 and 2 DOFS (depending on the surface and solar zenith angle). Given that 13CH4 is roughly 1.1 % of the total methane signal, we deemed it very unlikely that setting a  $(10\%)^2$  variance for 13CH4 would yield any total column information. We therefore decided to increase the magnitude of variance in order to establish the point when DOFS>1 can be achieved. We accept that such a method will drastically increase a priori and a posteriori errors, but we aim to reduce these through long term averaging.

We have included a discussion in the manuscript to this effect, Page.10, lines 26-31 and Page 11, lines 1-6.

*W.r.t to off-diagonal elements, a discussion on why they are necessary has been included.*

**Please see Page 11, lines 22-27.**

- page 12/13, Figure 2 and 3: The color plots are not clear, please use other color scale or representation.

*Could you please elaborate on why the figures are not clear? Reviewer 2 has identified all figures as being clear. Thank you.*

- page 17/18: It is mentioned that the combined band 2 and band 3 retrieval significantly increases computation time compared to band 2 or band 3 retrieval. Stating the CPU time for all cases would be useful to make that point. *The code/analysis method used in this paper was not an optimized retrieval*

The code/analysis method used in this paper was not an optimized retrieval code, but a linear analysis of the Averaging Kernels and a posteriori errors, incorporating the ORFM, and as such we do not feel that stating the exact CPU time for each analysis would be beneficial, since this will not relate to any fully optimized algorithm. However we have included a rough estimate of the time difference between considering each band individually, and combining them. This number is caveated with the facts stated above.

Please see Page 21, lines 26-29.

- I am missing a discussion on how methane isotolopogue retrievals, if successful, could be validated. Please include a discussion on possible validation strategies, e.g. using NOAA measurements.

Thank you for this important point, we have now included a discussion on potential future validation strategies, including NOAA, CTMs and TCCON.

Please see section 8, Page 26, lines 6-18.

**Technical corrections**:**

-"plant-based" or" plant based", use one consistently throughout the text *Thank you, we have changed this, and similar examples throughout the text.*

page 7, line 13: definite -> define
*Thank you, we have changed this.*

Page 9, line 25

- page 11: severally - > severely Thank you, this sentence was removed, rephrased and placed in section 2.2 in accordance with the recommendation of reviewer 2.

Please see Page 7, lines 1-3.

Other changes not specified in the above comments:

*With inclusion of a new Figure 1 and 5, all of the old figures have been renamed as appropriate, along with all references to the original figures.*

In response to a query from reviewer 2, we have inserted an example optical depth plot for 13CH4, in order to emphasize the limited optical depth of 13CH4. *Please see Figure 5 Page 19*,

Based on the modification of the target 13CH4 precision we have updated the results and conclusions shown in Section 7.1, 7.2 and 7.3, revising the target precisions, and the length of averaging times required. Please see, Page 24, lines 9-21, 28-33, Page 25, lines 1-6, 13-14.

Based on the inclusion of discussion points on the Philips-Tikhonov method, and future validation methods, the conclusions and summary section is now section 10.

Based on updates to the precision estimates, we have updates the metrics stated in the conclusions and summary section. Please see Page 27, lines 15-29 and Page 28, lines 1-6.

As above, we have updated all results stated in the abstract.

---

## Author Comment (AC2) · 20 Dec 2017

Dear reviewer, thank you for your review of our manuscript, and the comments below. In order to respond to your comments, we have kept your original comments in black non-italics. Our responses are in bold blue italics, and changes to the manuscript are in bold blue underlined italics.

Methane isotopologue detection is important for methane source detection. Measurements of isotopologues from satellite would be very important. The paper by Malina et al. is an interesting and informative paper on methane isotopologue detection capability, and is very relevant for GOSAT-2 and other future SWIR spectrometers. In general the story of the paper is well-written (although the text somewhat sloppy) and the figures are clear. The paper is suited for AMT. The paper can be accepted after the following comments are addressed.

**Main comments:**

- Sect. 2.1: A better description of the applicability of the limb sounding forward model ORFM for a nadir viewing instrument like GOSAT-2 is needed. There are missing processes in the ORFM model, like atmospheric scattering. Is surface reflection well included? Is surface elevation included?

Thank you for this point. We have expanded this section to give more details on how the ORFM can be applied to nadir viewing instruments, describing how surface reflectance and altitude is handled.

Manuscript adjusted in updated section 2.2, Page 6, lines 20-33 and Page 7 lines 1-3, 6,10-11.

- The term "solar inclination angle" is not used in nadir remote sensing. Therefore, this term should be converted to the term "solar zenith angle", which is 90 degrees – solar inclination angle. Please use the symbol \theta\_0 for the solar zenith angle and \theta for the viewing zenith angle.

We have changed all references to solar inclination angle through the manuscript to solar zenith angle. Satellite inclination angle has been changed to viewing zenith angle.

Note that all values in the original discussion paper were solar zenith angles, we had just mislabeled them, therefore none of the calculations have changed.

- P. 8, l. 17: What is the basis of setting the scaling factor f to (10 %)2 variance ? This means that you needed a much larger deviating delta13C than can be anticipated. What is the basis of that assumption?

This is an important point, raised by reviewer 1 as well. The reason for this is based on the relationship between the a priori covariance and the DOFS from the assumed GOSAT retrieval. From experience we know that methane covariance is often set to (10%)2 variance, in order to allow for some variation in the retrieved solution. At this level of variance we can expect between 1 and 2 DOFS (depending on the surface and solar zenith angle). Given that  ${}^{13}$ CH4 is roughly 1.1 % of the total methane signal, we deemed it very unlikely that setting a (10 %)2 variance for  ${}^{13}$ CH4 would yield any total column information. We therefore decided to increase the magnitude of variance in order to establish the point when DOFS>1 can be achieved. We accept that such a method will drastically increase a priori and a posteriori errors, but we aim to reduce these through long term averaging.

*We have included a discussion in the manuscript to this effect, Page.10, lines 26-31 and Page 11, lines 1-6.*

**Minor and Textual comments:**

- P. 2, l. 6: acronym GHG was already explained on the previous page *Thank you for spotting this, we have removed the acronym definition.*

Text "Greenhouse Gases" removed, Page 2, line 6.

- P. 2, l. 28: please give a reference for VPDB *We have inserted a reference for VPDB.*

**Page 2, line 28.**

P. 3, l. 18: 6ppbv: please add a space between the number and the unit. This holds throughout the paper, at many places, for many quantities, including %.
 Thank you for pointing this out, we have changed this wherever it appears in the manuscript.

- Table 1: please give the spectral resolution of the bands. *This has been included. Page 8, Table 1*

- Eq. 7: DOFS: acronyms should not be in italics because they are not symbols – *Thank you, this has been changed. Page 9, line 20.*

- P. 11, l. 4: please give an example of such errors. *Examples given, Page 14, line 5*

P. 11, l. 18: channel > channels
 *Thank you, this has been changed, Page 14, line 18*

- P. 11, l. 30-32: this should be mentioned earlier. *We have removed this statement in this position, and included a similar statement into Section 2.2.*

Please see Page 7, lines 1-3

- P. 12, l. 17: why a comma after 2\nu3? *Removed, Page 15, line 17*

- P. 13, l. 8: phenomenon > phenomena *Thank you, updated, Page 16, line 8*

- P. 14, l. 1-2: All simulations ..: please add this information to the main text because it is important information.

*These parts of the captions have been removed, and placed at the start of section 5. Please see, Page 15, lines 5-7.*

P. 14, l. 10: spectral irradiance > solar irradiance
 *Thank you, this has been changed.* Please see Page 17, line 10.

- At the same line: please remove: "due to blackbody solar emissions" (which is a strange comment)
  *Thank you, this has been changed. Please see Page 17, line 10.*
- P. 14, l. 20-23: too long sentence please make shorter sentences. *Thank you, this has been changed. Please see Page 17, lines 21,22.*
- P. 15, l. 6: At which wavelength do these albedo values hold? Please give a reference for these surface albedo values.

We have included a link to the ADAM albedo database, which gives a comprehensive review of global surface albedos based on MODIS data (http://adam.noveltis.com/). Please see Page 18, lines 6 and 7.

- P. 15, l. 17: please refer to Eq. 11 for the definition of f *Thank you, this has been inserted. Please see Page 18, line 19*

- P. 16, l. 9: why does the inclination angle not matter? This is unexpected. The air mass is much larger at smaller inclination angles.

We were surprised by this result as well, for the reasons you mention. However given how low the optical depth of 13CH4 is, it makes sense that varying the solar zenith angle will not make much difference. The optical depth of 13CH4 in band 2 of TANSO-FTS-2 is highlighted for two narrow regions in the figure below.

Figure 5. Optical depth covering two narrow  ${}^{13}CH_4$  spectral regions in band 2 of TANSO-FTS-2, the green line represents optical depth of all gases present in this portion of the spectrum (CH4, CO2 and H2O), whilst the blue line shows optical depth of purely the methane isotopologue  ${}^{13}CH_4$ : (a) indicates optical depth in the wavelength range 1658-1659 nm; (b) shows optical depth in the wavelength range 1670-1671 nm.

*We have included this Figure and detail in the manuscript, please see, Page 19, lines 10-19.*

- P. 16, l. 10: The hotspot depends on the scattering angle, not on the

inclination angle; the sun glint depends on viewing and solar geometry. *We have removed this statement, please see Page 20, line 1.*

P. 16, l. 11: extreme angles > special geometries
 Thank you, we have changed this statement as suggested, please see Page 20, line 2.

- P. 16, l. 23: " ..., this is an ...": please start a new sentence *Thank you, this has been changed. Please see Page 20, line 15.*

- P. 17, l. 8: . . . therefore: please start a new sentence *Thank you, this has been changed. Please see Page 20, line 31.*

- P. 17, l. 13: remove: including *Thank you, this has been changed.* *Please see Page 21, line 4.*

- P. 17, l. 33: manuscript > paper (also on next page) *Thank you, this has been changed.* *Please see Page 1, line 17, Page 21, line 24,27.*

- Caption fig. 4: Degrees of Freedom for Signal *Thank you, this has been changed. Please see Page 22, line 2.*

- Caption Table 4: Summarisation > Summary; and 6 surface albedos *Thank you, this has been changed. Please see Page 22, Table 4 caption.*

- Table 4: remove DOFS in the left-hand column since it is superfluous *Thank you, this has been changed. Please see Page 22,23, Table 4.*

- P. 19, l. 20: remove the points around below. *Thank you, this has been changed. Please see Page 23, line 20.*

- P. 20, l. 5: a priori should be in italics, and not in quotation marks (throughout the paper)

Thank you, this has been updated throughout this paper, along with any a posterioris showing similar properties.

- Caption Fig. 5: remove the second word retrieval *Thank you, this has been changed. Please see Page 26, line 3.*

- Caption Fig. A1: f should not be in quotation marks, since it is a normal symbol

All Figure captions which show 'f' have been updated.

- Figures: Fig. 1: please give the unit of the x-axis

Figure updated to include units on x-axis. Please see Figure 2, Page 13.

References:

- The references are very sloppy. First of all, the authors should replace the URLs by normal journal references. Please replace capital font for titles by normal font. Aydin et al.: the journal name is missing.

*References have been tided up as requested. Please see, Page 33 to end of Page 37.*

Other changes not specified in the above comments: With inclusion of a new Figure 1 and 5, all of the old figures have been renamed as appropriate, along with all references to the original figures.

Based on the modification of the target 13CH4 precision we have updated the results and conclusions shown in Section 7.1, 7.2 and 7.3, revising the target precisions, and the length of averaging times required. Please see, Page 24, lines 9-21, 28-33, Page 25, lines 1-6, 13-14.

Based on the inclusion of discussion points requested by reviewer 1 on the *Philips-Tikhonov method, and future validation methods, the conclusions and summary section is now section 10.*

Based on updates to the precision estimates, we have updates the metrics stated in the conclusions and summary section. Please see Page 27, lines 15-29 and Page 28, lines 1-6.

As above, we have updated all results stated in the abstract.

---

## Author Response (AR2)

**Dear Dr Stammes, thank you for your review of our manuscript, and the comments below. In order to respond to your comments, we have kept your original comments in black non-italics. Our responses are in bold blue italics, and changes to the manuscript are in bold blue underlined italics.**

5 Please check if the corrections due to reviewers' comments where appropriately inserted. I saw two strange points:

p. 6, l. 29: unclear sentence

*Thank you for indicating this issue, we have modified the sentence accordingly.*

10 *Please see P.6, lines 28-30*

p. 7, l. 11: incomplete sentence?

*Thank you for pointing this out, we have rephrased this sentence.*

*Please see P.7, lines 10-11*

[revised manuscript text omitted]

**2. Study Requirements, Models and Instruments**

In the following subsection, we describe the requirements for detecting methane isotopologues, and the tools and assumed instruments employed during the course of this research.

15  **2.1. Precision Requirements for Retrieval**

Analysis of global $\delta^{13}C$ concentrations by Nisbet et al. (2016) shows that trends and variations in $\delta^{13}C$ on a regional scale are of the order of a few per mil, which suggests that any total column retrieval algorithm will have to obtain better than this precision in order to comment on trends in $\delta^{13}C$. Given the above assessment, a much more likely prospect is the analysis of localised regions, where Nisbet et al. (2016) state that they can see wider ranges in the $\delta^{13}C$ of different source regions, for
20  example, Arctic and boreal wetland regions showing a per mil value of -70, while Siberian gas fields are at the -50 mark.

The $\delta^{13}C$ ratio is calculated as follows:

$$\delta^{13}C = \left( \frac{\left(\frac{13C}{12C}\right)\text{sample}}{\left(\frac{13C}{12C}\right)\text{standard}} - 1 \right) \times 1000, \quad (1)$$

25  where 'sample' refers to the current measurement and 'standard' refers to the VPDB value. Rigby et al (2012) suggest that there is a minimum margin of 10 per mil in terms of differentiating between fossil fuel and biogenic sources. Using this margin as a base, and applying Eq. (1), we can estimate the minimum precision on $^{13}C$ methane measurements required to achieve this per mil margin. Firstly we perform a trivial rearrangement of Eq (1) to make the sample $^{13}CH_4$ as the subject of the equation. We then specify two ranges of values in order to calculate the range of $^{13}CH_4$ values that will likely be
30  encountered terrestrially. 1) $\delta^{13}C$ range from -80 to -10 per mil, in 10 per mil steps; 2) $^{12}CH_4$ range calculated from a $CH_4$

range of 1770 – 1830 ppbv in 5 ppbv steps, where $^{12}CH_4$ is assumed to have been calculated from the HITRAN $^{12}CH_4$ abundance ratio (0.988274; see http://hitran.iao.ru/molecule/simlaunch?mol=6). Using these ranges, we can represent an expected range of terrestrial $^{13}CH_4$ abundances using Eq. (1).

[Figure]

Figure 1. Range of expected terrestrial $^{13}CH_4$ values (y-axis) given a range of $CH_4$ values between 1770 and 1830 ppbv, and $\delta^{13}C$ between -80 and -10 per mil (x-axis). The diagonal solid lines represent the $^{12}CH_4$ values for a given $CH_4$ value, while varying the $\delta^{13}C$ range. There are 13 $^{12}CH_4$ lines representing the $CH_4$ range in 5 ppbv steps. The red line (a) shows the $^{13}CH_4$ change between -50 and -40 $\delta^{13}C$ for a $CH_4$ of 1770 ppb; (b) is as (a), but includes a $CH_4$ change of 5 ppbv; (c) is as (a) and (b) but includes a $CH_4$ change of 15 ppbv.

Taking line (a) in Fig. 1, we show that a change in $^{13}CH_4$ of ~0.2 ppbv is observed for a 10 per mil $\delta^{13}C$ shift, this initially implies that a precision of 0.2 ppbv on $^{13}CH_4$ retrievals is required to resolve $\delta^{13}C$ measurements to a 10 per mil resolution. However, we have to take into account the precision on total column methane measurements from GOSAT, which Yoshida et al. (2011) state to be 6 ppbv on average, and roughly 15 ppbv at minimum (with Parker et al. (2015) also showing a minimum precision of roughly 15 ppbv). Taking these precisions into account, lines (b) and (c) in Fig. 1 show that there is an additional 0.053 ppbv uncertainty on $^{13}CH_4$ concentration for 5 ppbv uncertainty on $CH_4$ (multiplied by 0.988274 to get $^{12}CH_4$) concentration, and a 0.16 ppbv uncertainty on $^{13}CH_4$ concentration for 15 ppbv uncertainty on $CH_4$ concentration. Using Eq. (1) we can determine that a 0.053 ppbv $^{13}CH_4$ uncertainty equates to a $\delta^{13}C$ uncertainty of 2.7 per mil, and a 0.16 ppbv $^{13}CH_4$ uncertainty equates to a $\delta^{13}C$ uncertainty of 8 per mil. Therefore the goal of this research is to establish if GOSAT-2 can reach a target $^{13}CH_4$ precision of 0.2 ppbv, equating to a $\delta^{13}C$ precision of 10 per mil, under the caveat that on

average there will be a 2.7 per mil uncertainty associated with this value, and a maximum uncertainty of 8 per mil. For the average methane precision, source differentiation should still be possible, however when considering the lowest precision methane, then it will be likely that we lose the ability to differentiate between source types for certain, but it may still be possible if the measured $\delta^{13}C$ values are at the extreme end of the scale. Because of this, we will set additional $^{13}CH_4$

5    precision goals of 0.147 ppbv and 0.04 ppbv, in order to achieve the eventual desired $\delta^{13}C$ precision of 10 per mil.

Note that Fig. 1 shows a linear relationship between $\delta^{13}C$ quantities, and between $CH_4$ quantities, therefore although the above assessment focused on the 10 per mil range between -50 and -40, the assessment applies to any ranges on the figure.

In addition to measurement precision errors, numerous studies have suggested that a bias of ~5 ppbv can be expected on methane retrievals in GOSAT (Parker et al., 2015; Schepers et al., 2012; Yoshida et al., 2013). These biases are caused by

10   numerous effects (e.g. errors in the forward models), and are explored in more detail in the previously mentioned papers. The calculations above suggest that for a bias of 5 ppbv, a $\delta^{13}C$ bias of 2.7 per mil can be expected. However, we expect additional biases to appear on $^{13}CH_4$ retrievals, in addition to those on $^{12}CH_4$ retrievals. These are difficult to quantify exactly because the previously reported biases of $CH_4$ from GOSAT are based on comparisons with the Total Column Carbon Observing Network (TCCON; (Wunch et al., 2011)). No such measurements exist for $^{13}CH_4$ currently and so we cannot

15   estimate these biases.

**2.2. The Oxford Reference Forward Model (ORFM)**

The ORFM (Dudhia, 2017) is a General Line by Line (GENLN2) based Radiative Transfer Model (RTM) originally developed at the University of Oxford to provide reference spectral calculations, for the Michelson Interferometer for

20   Passive Atmospheric Sounding (MIPAS) instrument based on the ENVISAT satellite. The MIPAS instrument was a 'limb' viewing instrument, and as such the ORFM originally could only handle limb based calculation. However in the intervening years the ORFM has been updated significantly in order to be applicable to nadir viewing instruments. In accordance with these updates multiple viewing geometries are possible (allowing for multiple instrument viewing types such as balloons, aircraft or satellites), 1D and 2D atmospheres can be leveraged depending on the application of the user, and surface

25   elevation can be modelled, either through modifying a model atmosphere, or through setting the height of a ground based observer. Surface effects can be modelled, in the case of TIR through modification of surface temperature and emissivity, and in the case of SWIR through a specular reflectance model. However, for surface reflectance we deemed that specular reflectance was not sufficient to accurately model surface scattering. We therefore replaced this specular model with a Lambertian reflectance model (on the assumption that Lambertian reflectance accurately represents ground surface

30   reflectance (Yoshida et al., 2011)). The ORFM allows for the modelling of advanced spectral effects such as water vapour continuums and carbon dioxide line mixing, which can be modified by the user through look up tables.

The ORFM does include a key drawback, which is the lack of an atmospheric scattering mode. It does allow for absorption due to aerosols, but not scattering and can model Rayleigh scattering (as an absorption feature). In the context of this study

E M 29/12/2017 12:30

we judged this feature to be less important, since the calculation of the $\delta^{13}C$ metric will apply the 'proxy' effect to the simulated spectra, and largely negate any scattering effects. Towards this end 
[revised manuscript text omitted]
_4$. We accept that this is a very rough approximation, but it is effective estimating a covariance starting point for $^{13}CH_4$.

However, this variance represents a significant hurdle for $^{13}CH_4$ retrieval. Even *a priori* covariance for methane retrievals from GOSAT is often not this restrictive. Eguchi et al. (2010) show examples of methane covariance at this magnitude level, but this is based on high levels of climatology analysis at which point the total column methane retrieval is closer to the *a priori* than to the satellite retrieval. Therefore algorithm developers often allow more variance in their covariance matrices in order to allow for more variation in the retrieved solution. Even with a more relaxed covariance matrix the DOFS on

GOSAT retrievals are normally between 1 and 2. Given that [13]$CH_4$ is roughly 1.1 % of the methane signal, and that total column retrievals are highly sensitive to the covariance matricies. It seems very unlikely that setting a [13]$CH_4$ covariance matrix to values of $(3 \%)^2$ or even $(10 \%)^2$ would yield any DOFS in the total column. We therefore deemed it necessary to allow the covariance to vary more significantly than this, in order to establish the point where DOFS > 1, at the cost of increased *a priori* and *a posteriori* errors. Our assumption is that any retrieved [13]$CH_4$ variances can be averaged out, spatially and temporally. Therefore, this study initially assumes a $(10 \%)^2$ variance.

This study defines two forms of the matrix, firstly a pure diagonal covariance matrix based on the equation:

$$S_{a,ii} = \sigma_{a,i}^2 f^2, \qquad (11)$$

where $\mathbf{S}_{a,ii}$ is element $i,i$, (atmospheric layer) of a diagonal matrix, $\sigma_{a,i}$ is the standard deviation of element $i$ of the *a priori* vector, which in the case of this assessment is initially set at $(10 \%)^2$, and $f$ is a scaling factor designed to increase or decrease the standard deviation of the elements of the covariance matrix. This factor $f$ is designed to determine at what point the inherent instrument noise no longer has any influence on the retrieval. Because [13]$CH_4$ is present in minimal quantities in the atmosphere, it was deemed necessary to explore the effects of a non-diagonal covariance matrix, where the off-diagonal elements are calculated using the equation (Illingsworth et al., 2014):

$$S_{a,ij} = \sqrt{S_{a,ii}S_{a,jj}}\,exp\left(\frac{-(z_i-z_j)^2}{z_S^2}\right), \qquad (12)$$

where $\mathbf{S}_{a,ij}$ refers to a given off-diagonal element of layer $i,j$, $z_i$ is the altitude of element $i$, $z_j$ is the altitude of element $j$ and $z_S$ is the smoothing length, nominally set between 1 and 3 km. Off-diagonal elements describe the relationships between each of the pressure/altitude levels and are not always necessary in trace gas retrieval. This is especially true in cases such as $CO_2$, which are highly stable in the atmosphere, and such knowledge is not needed. However in the case of methane (and especially low concentration gas such as [13]$CH_4$) it is important to determine the pressure level effects, since this could lead to large changes. Including off-diagonal elements will increase algorithm computation time, but will likely result in a more accurate solution.

The other key gases present in bands 2 and 3 of GOSAT-2 ([12]$CH_4$, $CO_2$, $H_2O$ and CO) are all set at $(10 \%)^2$ of the MIPAS atmospheric profile, matching the initial value of the [13]$CH_4$ covariance matrix, with Herbin et al. (2013) suggesting similar variations at their peak.

[Figure]

**Figure 2.** *A priori* **gas concentration profiles of the main gases of interest.**

The MIPAS model assumes a total column averaged methane concentration of 1740 ppbv, which assuming a [13]C ratio of 1.1 % , equates to a total column averaged concentration of 19.14 ppbv for $^{13}CH_4$.

The information content analysis *a priori* setups, and simulation setups are summarised in the table below following the style of Herbin et al (2013):

**Table 2. Parameters for Information Content Analysis**

[revised manuscript text omitted]

10 The results shown in Table 4 suggest that solar zenith angle is not an important factor in retrieval for band 2 (in relation to $^{13}CH_4$ rather than methane). This is most likely because the optical depth of $^{13}CH_4$ is so low that changing the solar zenith angle does not change the $^{13}CH_4$ air mass significantly. In order to highlight this point, we include an ORFM simulation of optical depth for two narrow wavelength ranges in band 2 of TANSO-FTS-2, which clearly shows very low optical depth values.

[Figure]

**Figure 5. Optical depth covering two narrow $^{13}CH_4$ spectral regions in band 2 of TANSO-FTS-2, the green line represents optical depth of all gases present in this portion of the spectrum ($CH_4$, $CO_2$ and $H_2O$), whilst the blue line shows optical depth of purely the methane isotopologue $^{13}CH_4$: (a) indicates optical depth in the wavelength range 1658-1659 nm; (b) shows optical depth in the wavelength range 1670-1671 nm.**

[revised manuscript text omitted]

*Max DOFS for Surface Albedo*

| *Albedo* | **0.1** | **0.2** | **0.3** | **0.4** | **0.5** | **0.6** |
|----------|------|------|------|------|------|------|
| *Scenario 1* | 0.85 | 0.89 | 0.92 | 0.94 | 0.97 | 0.99 |
| *Scenario 2* | 0.85 | 0.89 | 0.92 | 0.94 | 0.97 | 0.99 |
| *Scenario 3* | 1.03 | 1.06 | 1.08 | 1.10 | 1.13 | 1.15 |
| *Scenario 4* | 1.06 | 1.09 | 1.13 | 1.16 | 1.19 | 1.22 |
| *Scenario 5* | 1.06 | 1.09 | 1.12 | 1.15 | 1.18 | 1.22 |

| | | | | | | |
|---|---|---|---|---|---|---|
| *Scenario 6* | 1.26 | 1.31 | 1.36 | 1.41 | 1.46 | 1.50 |
| *Scenario 7* | 1.08 | 1.12 | 1.16 | 1.19 | 1.23 | 1.26 |
| *Scenario 8* | 1.08 | 1.12 | 1.15 | 1.19 | 1.22 | 1.25 |
| *Scenario 9* | 1.30 | 1.35 | 1.41 | 1.46 | 1.51 | 1.55 |

**7. Error Analysis**

Even if sufficient DOFS can be established to identify where $^{13}CH_4$ retrievals are influenced more by the measurement than by the *a priori*, the errors associated with the retrieval may well make identifying methane source types a practical impossibility. Therefore an assessment of the expected total column errors is required, these errors for $^{13}CH_4$ can be summarised as (Yoshida et al., 2011):

$$\sigma = \frac{\sqrt{h^T S h}}{h^T 1}, \quad (16)$$

$$h^T = \left( w_{dry,1} w_{dry,2} \dots w_{dry,n} \right), \quad (17)$$

where $\sigma$ is the total column *a posteriori* error, depending on the subset of altitudes or pressures used, $h$ is the dry air partial column, calculated from $n$ layers of the retrieval grid (21 in this case) where $w_{dry}$ is calculated from the model pressure profile and $H_2O$ concentration profile. $1$ is a column vector with elements of unity and a length equivalent to the dry air partial column.

Using the scenarios outlined in Table 3, the total column error (along with the interference, smoothing and measurement errors) for $^{13}CH_4$ retrieval can be established. Total column errors for $^{12}CH_4$ are assumed to be documented in studies such as Parker et al. (2011) and Yoshida et al. (2011).

Results from scenarios 1, 4 and 7 are shown in Fig. 7 below. The results from scenarios 2, 5 and 8 are shown in Fig A3. and scenarios 3, 6 and 9 are shown in Fig. A4 both of which are in the Appendix.

**7.1. Band 2**

Based on the summation of errors identified above, we can estimate the precision of a synthetic retrieval in band 2 of TANSO-FTS-2, for the range of *a priori* covariance matrices identified previously.

Scenario 1 (Fig. 6(a)) suggests that retrievals are heavily biased towards the *a priori* in band 2 of TANSO-FTS, except perhaps for a variance of $(100\ \%)^2$, over a very bright surface (i.e. albedo of 0.6, or SNR equal to 500). In this case, the maximum precision for a single sounding equates to 2.4 ppbv for $^{13}CH_4$. Based on the total column averaged concentration of $^{13}CH_4$ from the MIPAS profile identified in Sect. 4.1, 2.4 ppbv precision equates to roughly 13 % error. For reference,

5    Yoshida et al. (2011) show that the average total column precision for methane retrievals is 5.86 ppbv, which equates to 3.4 %. Based on the DOFS values for scenario 2 (Fig. A1(a)), we can assume similar precision values, however, the DOFS values for scenario 3 (Fig. A2(a)) suggest that unity is achieved for variance of $(80\ \%)^2$ or above for the whole SNR range. Based on the error values shown in Fig. A4(a), the maximum precision at this variance is 2.2 ppbv, and the minimum is 3 ppbv, equating to 11.5 % error and 15.6 % respectively. These results are far from the base 0.2 ppbv precision requirement

10   set in sect. 2.2 (and especially far from the modified targets of 0.147 ppbv and 0.04 ppbv), however, the precision can be increased through averaging multiple retrievals together (both temporally and spatially), where the standard deviation is inversely proportional to the square root of the number of measurements (Parker et al., 2015). Therefore for scenario 3, we suggest that a precision of 0.2 ppbv can be achieved using the average of 121 measurements for high SNR, and 225 measurements for low SNR, both of which are achievable over a significant period of time with a large spatial region for

15   GOSAT-2. Taking into account $^{12}CH_4$ uncertainty, which increases the $^{13}CH_4$ precision requirements, we assume that when averaged over the numbers of measurements identified above we can expected a $^{12}CH_4$ precision of 5 ppbv, equating to a required increase in $^{13}CH_4$ precision to 0.147 ppbv which can be achieved using the average of 224 measurements for high SNR, and 416 measurements for low SNR. These scales of measurements seem less likely to be achievable with the explicit goal of determining source types, however it may be possible to investigate globally averaged temporal trend climatology or

20   hemispherical biases. Note that these metrics do not include the effects of any potential biases in the retrievals, which will not be removed through spatial and temporal averaging and can potential skew the assumed $\delta^{13}C$ values.

**7.2. Band 3**

Using the methods identified above in band 2, the total errors for band 3 retrievals are explored.

25   Scenario 4 (Fig. 6(b)) shows that unity DOFS occurs for variance of $(80\ \%)^2$ or above for the whole SNR range. Using Fig. 7(b), we can suggest that the maximum and minimum precision is 1.5 ppbv and 1.8 ppbv respectively, these equate to 7.8 % and 9.4 % of the total column. We suggest that through spatial and temporal averaging, these errors can be reduced to the 0.2 ppbv target by averaging 56 and 81 measurements for maximum and minimum SNR cases, and in the case of methane precision errors of 5 ppbv, (104 and 150 measurements) and 15 ppbv (1400 and 2025 measurements). Changing the solar

30   zenith angle for scenario 5 (Fig. A1(b)) does not impact the DOFS significantly, however, if we consider scenario 6 (Fig. A2(b)), DOFS of unity occur for variance of $(40\ \%)^2$ or above for the whole SNR range. The maximum and minimum precisions at this variance are 1.1 ppbv and 1.3 ppbv (Fig. A4(b)) respectively. The target precision can be increased through averaging 20 and 28 measurements respectively, and in the case of $^{12}CH_4$ precision errors of 5 ppbv, (56 and 78

measurements) and 15 ppbv (756 and 1050 measurements). Therefore for retrievals with band 3, we suggest that measurements of $\delta^{13}$C to an accuracy of 10 per mil can be achieved within monthly periods of GOSAT-2 measurements, assuming small levels of $^{12}$CH$_4$ precision errors, which are achievable when averaged over large volumes of data. If we assume that there is a constant 5 ppbv or greater $^{12}$CH$_4$ precision errors, then we suggest that $\delta^{13}$C for Transcom region scale analyses are more appropriate (Takagi et al., 2014). Again the potential for biases caused by systematic errors must be considered, with section 2 suggesting that a minimum $\delta^{13}$C bias of 5 per mil can occur.

**7.3. Combined Band 2 and Band 3**

Scenario 7 (Fig. 6(c)) shows that unity DOFS occurs for variance of (70 %)$^2$ or above for the whole SNR range. Using Fig. 7(c), we suggest that the maximum and minimum precision at this variance is 1.5 ppbv and 1.8 ppbv respectively, i.e. very similar to those found in band 3 scenario 4, therefore similar numbers of measurements are required in order to achieve the desired precision. If we consider scenario 9, unity DOFS are achieved for a variance of (35 %)$^2$ (Fig. A2(c)), the maximum and minimum precisions at this variance are 0.7 ppbv and 1.2 ppbv (Fig. A4(c)) respectively. The target precision of 0.2 ppbv can be increased through averaging 12 and 36 measurements respectively, and in the cases of $^{12}$CH$_4$ precision errors of 5 ppbv, (23 and 67 measurements) and 15 ppbv (306 and 900 measurements). Scenario 9 shows the best results in terms of information content, and measurement precision, to the point where over highly reflective surfaces very few measurements are required in order to make an accurate assessment of the source type.

[Figure]

**Figure 7. Synthetic total column retrieval errors for scenarios outlined in Table 3, based on Eqs. (8,9,10 & 16). The maximum and minimum values (based on maximum and minimum SNR), for interference, measurement and smoothing error are shown, in addition to the total error: a) indicates scenario 1, b) indicates scenario 4, c) indicates scenario 7.**

**8. Potential Validation Methods**

The next step to performing retrievals of $^{13}CH_4$ from GOSAT-2 is validating these measurements. This is currently a challenging topic since there are currently no total column measurements of $^{13}CH_4$ in the public domain. As discussed in the introduction, the only currently available measurements are available from NOAA flask data (Nisbet et al., 2016), land or airborne surveys of specific locations (Fisher et al., 2017), stratospheric balloon measurements (Röckmann et al., 2011), or ACE-FTS limb measurements (Buzan et al., 2016). These measurements only cover specific sections of the atmosphere and cannot be directly compared to any total column measurements. Having said this, comparisons can be made, under the caveat that biases will exist between the measurement techniques, due to atmospheric circulation and/or fractionation. Studies that attempt this having adequately described what these biases could be would be a major step forward.

Another potential avenue is to modify currently existing global Chemistry Transport Models (CTMs) to incorporate $^{13}CH_4$ transport and fractionation, based on surface measurements from NOAA flask data, this method could adequately represent total column $^{13}CH_4$. Buzan et al. (2016) attempt this with the Whole Atmosphere Community Climate Model (WACCM) in order to compare against ACE-FTS measurements, with mixed results.

Finally the TCCON network mentioned above is perhaps the most useful avenue for pursuit. Although $^{13}CH_4$ measurements are not currently available from TCCON, some minor modifications to the standard GGG2014 algorithm should provide the appropriate utility. TCCON has spectral sensitivity to band 2 of TANSO-FTS-2, and because TCCON retrievals are not dependent of solar backscatter, can obtain much higher SNR measurements.

5  **9. Alternatives to *a priori* Methods**

This study has been performed largely with the JAXA/NIES/MOE GOSAT retrieval algorithm (Yoshida et al., 2011; 2013) in mind. With the hope that only minor modifications to the algorithm will allow for retrievals of $^{13}CH_4$ from GOSAT-2. This of course extends to any other *a priori* based retrieval algorithm. However there is an important argument to be had about the usefulness of *a priori* methods in the case of $^{13}CH_4$, when such huge covariances are required in order to obtain

10  information content. The operational methane algorithm on the recently launched TROPOMI uses the Philips-Tikhonov regularisation scheme (Hu et al., 2016) which makes use of a regularisation parameter instead of *a priori* data or covariances. Because currently available schemes do not easy provide values for $^{13}CH_4$ *a priori* data, the Philips-Tikhonov may be a more suitable method for future algorithm development.

**10. Conclusions and Summary**

15  To summarise, this work investigates the possibility of whether $^{13}CH_4$ can be retrieved with a sufficient level of accuracy by bands 2 and 3 of the GOSAT-2/TANSO-FTS-2 instrument, in order to make a judgement on the nature of a methane source type (biogenic, thermogenic or abiogenic), via the use of the $\delta^{13}C$ ratio. We assume that an accuracy of 10 per mil of $\delta^{13}C$ values is sufficient to distinguish between methane source types, as shown by Rigby et al. (2012), and with this accuracy, we calculate that a minimum $^{13}CH_4$ retrieval precision of 0.2 ppbv is required in order to achieve $\delta^{13}C$ with a 10 per mil

20  accuracy, but preferably 0.147 ppbv or 0.04 ppbv when taking into account precision errors on $^{12}CH_4$.
Using the well established DOFS methods (Rodgers, 2000) the RTM ORFM, and the assumption of clear sky conditions we calculate the key metrics of DOFS and total retrieval error in order to judge a) the information content in a retrieval, and b) the precision of that retrieval, based on a series of test *a priori* covariance matrices. Using a combination of bands 2 and 3, we find that total column retrieval of $^{13}CH_4$ with sufficient DOFS is possible, with a maximum and minimum precision of

25  0.7 ppbv and 1.2 ppbv respectively. Assuming statistical error reduction techniques, this precision can be increased to 0.2 ppbv by averaging over 12 and 36 measurements respectively, to 0.147 ppbv by averaging over 23 and 67 measurements respectively and to 0.04 ppbv by averaging over 306 and 900 measurements respectively. This number of measurements for the best two target precisions is certainly achievable over a monthly period, assuming modest spatial sampling of 2°x2°, which is often how GOSAT data is represented. Implying that GOSAT-2 will be able to differentiate between methane

source types at a high temporal resolution. However in the case of high precision errors on $^{12}CH_4$, representation of $\delta^{13}C$ on Transcom regional scales is a more feasible prospect.

This analysis was also applied to bands 2 and 3 individually, and it was found that band 2 can achieve enough DOFS for a $^{13}CH_4$ retrieval at the desired precision, based on averaging up to 225 measurements for a completely unconstrained *a priori* covariance matrix. Band 3 showed similar results to bands 2 and 3 combined but required up to 81 measurements in order to achieve the required precision. These are in the cases of no or limited $^{12}CH_4$ precision error.

Across all bands, we find that the DOFS and precision are significantly affected by the instrument SNR, but not by the solar zenith angle to any significant degree. In addition, the rate of increase of DOFS with respect to scaling factor is significantly higher in the combined bands, than either band when considered individually. However, combining the DOFS from both bands leads to a significant computational penalty.

Given the relative abundance of $^{13}CH_4$ spectral lines in band 3 of GOSAT-2, there is also some scope for future comparisons with measurements from Sentinel 5/5-P, both of which are sensitive to the same spectral regions. In addition, it is envisaged that there will be a period where GOSAT and GOSAT-2 are in simultaneous operation, and it may be possible to combine the measurements from both of these satellites in order to reduce uncertainty in isotopologue measurements.

**11. Appendices**

The scenario plots not indicated in the main text are shown below, namely scenarios 2, 3, 5, 6, 8 and 9 for both DOFS and retrieval errors.

[Figure]

**Figure A1. Degrees of Freedom for $^{13}CH_4$ vs scaling factor $f$ for scenarios outlined in Table 3. Each coloured line represents different surface albedo conditions as shown in the key. The black dashed line represents unity DOFS: (a) indicates scenario 2, (b) indicates scenario 5, (c) indicates scenario 8.**

[Figure]

**Figure A2. Degrees of Freedom for $^{13}CH_4$ vs scaling factor $f$ for scenarios outlined in Table 3. Each coloured line represents different surface albedo conditions as shown in the key. The black dashed line represents unity DOFS: (a) indicates scenario 3, (b) indicates scenario 6, (c) indicates scenario 9.**

[Figure]

**Figure A3.** Synthetic total column retrieval errors retrieval for scenarios outlined in Table 3, based on Eqs. (8,9,10 & 16). The maximum and minimum values (based on maximum and minimum SNR), for interference, measurement and smoothing error are shown, in addition to the total error: a) indicates scenario 2, b) indicates scenario 5, c) indicates scenario 8.

[Figure]

**Figure A4. Synthetic total column retrieval errors retrieval for scenarios outlined in Table 3, based on Eqs. (8,9,10 & 16). The maximum and minimum values (based on maximum and minimum SNR), for interference, measurement and smoothing error are shown, in addition to the total error: a) indicates scenario 3, b) indicates scenario 6, c) indicates scenario 9.**

**Author Contribution**

E.M., Y.Y., M.T. and J.-P.M. conceived and designed the experiments; E.M. performed the experiments, analysed the data and wrote the paper.

**Acknowledgements**

This research was partially funded by NIES GOSAT-2 Project, in combination with the first author's PhD grant from the National Centre for Earth Observation (NCEO) through the Natural Environment Research Council (NERC) based in the UK (award number 157550). We would also like to acknowledge Anu Dudhia at Oxford University for the ORFM, and JAXA/NIES/MOE for GOSAT-TANSO-FTS data.